# Compaction effects on evaporation and salt precipitation in drying porous media

**Nurit Goldberg-Yehuda**[1,2], **Shmuel Assouline**[1], **Yair Mau**[2], **and Uri Nachshon**[1]

[1]Institute of Soil, Water and Environmental Sciences, Agricultural Research Organization – Volcani Institute, Rishon Lezion 7505101, Israel
[2]The Institute of Environmental Sciences, The Robert H. Smith Faculty of Agriculture, Food and Environment, The Hebrew University of Jerusalem, Rehovot 7610001, Israel

**Correspondence:** Uri Nachshon (urina@agri.gov.il)

**Abstract.** Compaction and salinization of soils reduce croplands fertility, affect natural ecosystems, and are major concerns worldwide. Soil compaction alters soil structure and affects the soil's hydraulic properties, and it therefore may have
5 a significant impact on evaporation and solute transport processes in the soil. In this work, we investigated the combined processes of soil compaction, bare soil evaporation, and salt precipitation. X-ray computed microtomography techniques were used to study the geometrical soil pore and grain param-
10 eters influenced by compaction. The impact of compaction on evaporation and salt precipitation was studied using column experiments. We found that compaction reduced the average grain size and increased the number of grains, due to the crushing of the grains and their translocation within the
15 compacted soil profile. Changes in pore and grain geometry and size were heterogeneously distributed throughout the soil profile, with changes most apparent near the source of compaction, in our case, at the soil surface. The column experiments showed that the presence of small pores in the upper
20 layer of the compacted soil profile leads to higher evaporation loss and salt precipitation rates, due to the increase of hydraulic connectivity to the soil surface and the prolongation of the first stage of evaporation.

## 1 Introduction

25 Soil compaction is a major cause of soil degradation in agricultural environments (Akker and Canarache, 2001; Hamza and Anderson, 2005; Pagliai et al., 2003). It is associated with the increase of soil bulk density and decrease of porosity (Mossadeghi-Björklund et al., 2016). Soil compaction at different intensities may occur due to natural processes, such 30 as the impact of raindrops, natural soil-forming processes, and animal treading, and to processes linked to human activities, especially in agricultural environments, such as intense vehicular traffic over the fields (Assouline, 2004; Pagliai et al., 2003; Shah et al., 2017; Mossadeghi-Björklund et al., 35 2018). Passing of heavy machinery and vehicles over the fields leads to compaction as a result of pure static stresses, wheel slip, and dynamic forces, caused by vibration of the engine and the attached implements (Horn et al., 1995). Barik et al. (2014) found significant variability in the spatial distri- 40 bution of the aggregate stability, bulk density, total porosity, penetration resistance, and moisture content values, following traffic operation over arable lands. Pores nearby the location of compression are strongly affected, whereas those located further away from the source of compaction are less 45 affected (Keller et al., 2019; Schlüter and Vogel, 2016). Thus, compaction resulting from traffic generally presents a sigmoidal distribution with depth of the soil bulk density, where the denser part is close to the surface (Assouline, 2004; Augeard et al., 2007; Bresson et al., 2004; Dejong-Hughes 50 et al., 2001; Horton et al., 1994; Reicosky et al., 1981).

To overcome soil compaction in arable lands and to loosen up the soil upper layer, soil tillage is implemented, producing favorable conditions for seed germination and crop root development. Soil tillage reduces soil bulk density, increases 55 porosity, homogenizes soil-wetting processes, and improves soil aeration in the root zone (de Almeida et al., 2018; As-

souline et al., 2014; Rasmussen, 1999; El Titi, 2003). On the other hand, under certain conditions, tillage may accelerate processes of soil erosion and compaction of the soil at the lower boundary of the tilled zone (de Almeida et al., 2018), i.e., the plough pan (Podder et al., 2012). Therefore, minimizing vehicle passing over the fields, reducing tillage, and improving our understanding of the nature of soil compaction is important.

## 1.1 Soil compaction at the microscale

At the microscale, stresses in soil do not propagate homogeneously but rather through preferential paths – in all directions (Nawaz et al., 2013). Consequently, soil deformation occurs at specific sites, where the stresses and strains are maximal. These areas, also known as shear bands, are those that undergo structural deformation, while soil volumes between the stress chains may preserve their original structure and porosity (Nawaz et al., 2013; Naveed et al., 2016). The heterogeneous effect of compaction on the physical properties of the soil leads to an uneven spatial distribution of the soil's hydraulic properties that in turn affect water flow and solute transport processes in the soil profile (Alaoui et al., 2018; Assouline, 2006a, b; Assouline and Or, 2006).

Soil compaction affects the pore network in the soil profile, with respect to (i) pore-size distribution, (ii) pore geometry and morphology, and (iii) pore connectivity (Horn et al., 1995; Mossadeghi-Björklund et al., 2016). Consequently, water-related soil properties are significantly altered (Horn et al., 1995; Assouline, 2006a, b). These changes affect unsaturated soil hydraulic properties and reduce saturated soil hydraulic conductivity, thus increasing surface runoff and soil erosion by water (Alaoui et al., 2011; Keller et al., 2013; Shah et al., 2017; Soane and van Ouwerkerk, 1995). In addition, soil aeration is reduced, and the heterogeneous changes of the soil's physical and hydrological properties may lead to the formation of preferential water flow paths in the soil (Alaoui et al., 2011; Keller et al., 2013; Soane and van Ouwerkerk, 1995). Consequently, solute transport and accumulation in the soil may be affected, impacting nutrient availability to plants (Horn et al., 1995; Lipiec and Stępniewski, 1995; Hendrickx and Flury, 2001; Mossadeghi-Björklund et al., 2016) The abovementioned changes of the soil properties, due to compaction, usually occur at the top 30 cm of the soil profile (Horn et al., 1995; Keller et al., 2019). These changes in the soil structure of the upper soil layer have impacts on the soil water balance in general and on infiltration and evaporation processes in particular (Assouline et al., 2007, 2014; Shokri et al., 2010; Sillon et al., 2003).

## 1.2 Bare soil evaporation

Evaporation plays a central role in the hydrologic cycle and surface energy balance (Bergstad et al., 2018), as it is the main process of soil-water transfer to the atmosphere (Brut-saert, 2005; Hillel, 1982). The evaporation in porous media is affected by and involves complex and highly dynamic interactions between boundary conditions, liquid flow, and vapor diffusion (Lehmann et al., 2008; Or et al., 2013; Assouline et al., 2014; Kamai and Assouline, 2018; Assouline and Kamai, 2019).

The evaporation process from bare soils consists of two stages: stage 1 (S1) and stage 2 (S2). Evaporation during S1 takes place at the soil surface, and a hydraulic connection is maintained throughout the soil profile, by capillary flow of water through the soil's small pores (Lehmann et al., 2008; Nachshon et al., 2011a, b; Bergstad et al., 2018; Assouline and Narkis, 2019). In parallel to the upward capillary flow, through the small pores, the larger pores in the soil are air invaded. The interface between saturated and partially dry regions is defined as the drying front (Shokri et al., 2008). S1 is characterized by a high and relatively constant evaporation rate affected by soil properties and atmospheric conditions (Hillel, 1982). S2 begins when a characteristic capillary head, $\psi_c$, is reached at the soil surface (thus the small pores are air invaded) and the hydraulic connection between the soil profile and the surface is lost CE1 (Prat, 2002; Lehmann et al., 2008; Assouline et al., 2014). The evaporation front, i.e., the upper boundary of the capillary rise through the small pores (Shokri et al., 2008), migrates downward, and evaporation rate is drastically reduced as vapor diffusion from the evaporation front to the atmosphere governs the process (Lehmann et al., 2008; Nachshon et al., 2011b; Or et al., 2013; Kamai and Assouline, 2018).

Over recent years, several studies have shown that soil structure has a strong effect on bare soil evaporation. Lehmann et al. (2008) and following studies (e.g., Lehmann and Or, 2009; Nachshon et al., 2011a, b) have shown that heterogeneous structure of the porous media, consisting of two texturally different matrices (coarse and fine) separated by a sharp interface perpendicular to the evaporation front, results in elongation of S1 and increased cumulative evaporation. In short, this is a result of the large pores of the coarse media that are being invaded by air much before the fine pores, with the lower (more negative) air entry pressure. The pressure head differences between the large and fine pores results in the effect that the coarse-texture domain supplies water, by capillary flow, to the fine-texture domain; thus, more water is available for S1 through the fine pores (Lehmann and Or, 2009).

In addition, structural changes of the soil along the vertical axis (with depth) may also affect evaporation (e.g., Or et al., 2007; Lehmann et al., 2008; Shokri et al., 2010; Assouline et al., 2014; Assouline and Narkis, 2019). It was shown that porous media composed of a fine-texture domain that overlies a coarse-texture domain may result in longer duration of S1 and increased cumulative evaporation with respect to the homogeneous domain, composed of the coarse-texture matrix only. In the layered structure, as soon as the drying front reaches the layers with the relatively larger pores, rapid wa-

ter displacement will occur from the large pores to the overlying finer pores. The pressure in the coarse layer changes abruptly from its air-entry value to the air-entry value at the evaporation front, which is associated with the higher capillary suction of the small pores (Or et al., 2007; Shokri et al., 2010). Consequently, the coarse-texture layer acts as a water reservoir that supplies extra water to sustain a longer S1 and higher cumulative evaporation compared to the homogeneous soil structure. It is important to emphasize that this process will occur only if the thickness of the fine-texture layer is shorter than its characteristic length as only at this state the drying front may reach the coarse-texture domain, while the system is at S1 and the evaporation front is still at the soil surface (Assouline et al., 2014; Assouline and Narkis, 2019).

## 1.3 Evaporation and soil salinization

Evaporation and soil salinization are tightly connected processes, especially in cultivated fields. Soil salinization in cultivated fields is a common feature resulting from low-quality irrigation water, fertilization, and saline and shallow groundwater resulting from inadequate irrigation and drainage practices (Yakirevich et al., 2013; Berezniak et al., 2018; Nachshon, 2018; Hopmans et al., 2021).

The presence of salts in the soil pore water reduces the osmotic potential of the solution and the equilibrium water vapor pressure (Nassar and Horton, 1997). Consequently, evaporation rates from a saline soil are expected to be lower compared to solute-free conditions. During evaporation, the concentration of the dissolved ions increases in the pore solution, until saturation is reached and salt precipitation begins (Nachshon et al., 2011a). Salt precipitation at the soil surface occurs mainly during S1, where the evaporation rate is maximal and solutes are continuously transported to the evaporation front at the soil surface by capillary flow. As the salt begins to precipitate and expands over the soil surface, the evaporation rate is affected by the pore-scale dynamics of the precipitated salt (Bergstad et al., 2017, 2018), and the consequent changes to liquid and vapor flow processes through the salt crust. The presence of porous media heterogeneities (Lehmann and Or, 2009; Nachshon et al., 2011a), initial solute concentration of the pore water (Rad and Shokri, 2012; Shokri-Kuehni et al., 2017a), soil surface properties (Nachshon et al., 2011b), and salt type (Shokri-Kuehni et al., 2017a) may affect the dynamics of the salt precipitation layer and its influence on evaporation (Bergstad et al., 2018). In some cases, if the precipitated salt layer over the soil surface is hydraulically connected to the solution in the pores below, it may accelerate evaporation, as the surface area of the precipitated salt is usually higher compared to the underlying bare soil. Consequently, as long as the salt crust can pump liquid water from the underlying media, the elevated surface area of the salt crust would increase total evaporation (Shokri-Kuehni et al., 2017b). In addition, roughness

changes of the matrix–atmosphere interface by the precipitated salt crust may also increase evaporation due to changes of wind speed and surface energy balance (Kampf et al., 2005; Nield et al., 2015). On the other hand, if the precipitated salt layer is hydraulically disconnected from the solution in the pores, it acts as a barrier that reduces vapor diffusion from the soil to the atmosphere, and cumulative evaporation and evaporation rates will be reduced (Nachshon et al., 2011a).

Previous studies have shown that changes in soil structure, which affect evaporation, also influence the nature and location of salt precipitation in the presence of saline solution (Bergstad et al., 2017; Nachshon et al., 2011a, b). The drying patterns and dynamics are greatly influenced by the presence of textural discontinuities that may result in preferential drying and promotion of capillary exchange between different regions in the soil (Bergstad et al., 2017; Lehmann and Or, 2009). As previously mentioned, soil compaction affects soil structural and textural properties, mainly at the soil surface, where evaporation and salt precipitation are prominent.

## 1.4 Evaporation and soil compaction

Studies on the effect of soil compaction on evaporation, in general, and its relation to salt precipitation, in particular, are scarce. Nassar and Horton (1999) examined salinity and compaction effects on soil water evaporation from bare soils, focusing on water and solute distributions in the soil. They showed that compaction increases cumulative evaporation, due to increased matric suction of the compacted soil, resulting in the increase of the soil water-holding capacity and unsaturated hydraulic conductivity. Consequently, water flows more efficiently from deep parts of the soil profile to the soil surface, where evaporation is maximal, at S1 evaporation. In their study, Nassar and Horton (1999) deliberately compacted the soil samples in a homogeneous manner, ignoring the heterogeneous nature of soil compaction. Moreover, while the authors examined the impact of compaction and evaporation on solute distribution in the soil profile, and its impact on the solution osmotic potential, they did not consider the interactions between soil compaction, evaporation, and salt precipitation.

Sillon et al. (2003), using indirect measurements under non-saline field conditions, also pointed at higher evaporation from compacted soils. The authors showed that for compacted soils, soil drying occurred from bottom to top, as opposed to regular evaporative conditions, where the drying front recedes from the surface downward. In agreement with Nassar and Horton (1999), this was explained by the high capillary suction of the compacted soil that enabled pumping of water from the lower parts of the soil profile to the soil surface, where evaporation takes place. Assouline and Narkis (2019) used a constructed multilayered porous medium, where the top layer had the highest bulk density, smallest grains, and smallest pores, and where the bulk den-

sity gradually decreased, while grains and pore sizes gradually increased in the underlying layers. They measured evaporation from this structure and from a structure where the order of the layers was reversed. It was shown that in the soil structure where the top layer had the highest bulk density the S1 duration was extended and the cumulative evaporation increased in comparison to the reversed structure. The concept of the characteristic length was applied to explain these results, providing a physically based support to the observations of Sillon et al. (2003).

The main objective of the work presented herein is to understand the impact of soil compaction on soil evaporation, solute distribution, and salt precipitation, as well as their interactions, along the soil profile. Relying on previous studies (Assouline and Narkis, 2019; Nassar and Horton, 1999; Sillon et al., 2003), we conducted a series of experiments to fill up the knowledge gaps regarding the complex interactions between the heterogeneous structural nature of compacted soils, evaporation, and salt dynamics.

## 2   Conceptual model

Based on the studies detailed above, we hypothesize that compacted soil may be considered a semi-layered structure where pore openings are minimal at the soil surface, due to compaction, and gradually increase with depth (Fig. 1a). Consequently, soil bulk density, capillary suction, water-holding capacity, and unsaturated hydraulic conductivity are maximal at the upper layer of the soil profile, as well as its characteristic capillary length. These structural changes will result in evaporation patterns similar to those observed for the layered structure domain where fine media overlie a coarser-texture domain.

It is hypothesized that for compacted conditions the first tip of the drying front will invade the underlying larger pores, acting as an air conduit that will allow air to replace the water that will be pumped upward by the fine-texture horizons (Fig. 1b2). Consequently, at the compacted soil scenario, during evaporation, the upper layers of the medium will retain high levels of saturation, while the matrix will be dried from bottom to top. This will lead to higher cumulative evaporation and longer S1 duration compared to the non-compacted state.

Under saline conditions, where the pores are filled with a salty solution, evaporation will lead to solute precipitation at the soil surface and to the formation of an efflorescent salt crust, at least in the case of NaCl (Nachshon and Weisbrod, 2015; Piotrowski et al., 2020). Under non-compacted conditions, the receding drying front during evaporation and the resulting increased matric potential and reduction of the soil water content near the soil surface will result in a quick transition into a state of a hydraulic discontinuity between the soil and the salt crust. Therefore, the salt crust will reduce evaporation, as it acts as a barrier for water vapor diffusion

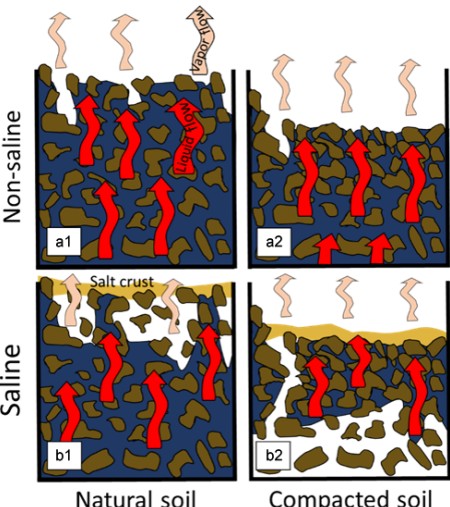

**Figure 1.** Conceptual model of evaporation and salt precipitation under compacted and non-compacted conditions; **(a)** initial stages of S1 evaporation with the first air invasion into the matrix; **(b)** advanced stages of evaporation, where most of the soil is hydraulically disconnected from the salt crust for the non-compacted state **(b1)**, whereas at the compacted state most of the soil is hydraulically connected to the salt crust; thus, evaporation front is at the salt crust upper surface **(b2)**. Drying of the compacted media is from bottom to top, as liquid water is replaced by air that is flowing downward through the larger pores that act as air conduits.

from the evaporation front to the atmosphere (Fig. 1b1). On the other hand, under the compacted soil condition, it is hypothesized that the drying pattern from bottom to top and the high water saturation that is sustained near the soil surface will maintain a hydraulic connection between the soil profile and the upper surface of the salt crust, which is now the active evaporation front of the system (Fig. 1b2). Therefore, it is hypothesized that even though more salt is expected to accumulate on the surface of the compacted soil, due to the higher evaporation, its impact on evaporation will be moderate compared to homogeneous non-compacted conditions.

These hypotheses were tested herein by means of high-resolution computed micro-tomography (µCT) scans, to characterize, at the microscale, the impact of compaction on soil pores and grains properties, and column experiments, to study the impact of compaction, at the macroscale, on evaporation and salt precipitation. A simple numerical model was also used to strengthen some aspects of the experimental findings and research hypotheses.

## 3   Materials and method

This study relies on three parts: (i) studying the impact of compaction on soil physical properties at the microscale and macroscale; (ii) simple numerical simulation of evaporation from different soil structures, mimicking compacted and

non-compacted conditions, with and without the presence of a precipitated salt crust; and (iii) validation of the conceptual model presented in Fig. 1 by means of macroscale evaporation experiments.

## 3.1 Microscale changes of pores and grains physical properties due to compaction

Imaging of sand samples before and after compaction was conducted in a non-destructive manner using a high-resolution µCT (SKYSCAN 1172, Bruker, Kontich, Belgium), in order to observe the impact of compaction on the sand physical structure, pore and grain dimensions, and spatial arrangement. The X-ray source voltage was 80 kV, and the electrical current was 10 mA TS1. The scan was done with aluminum and copper filters, with an image rotation of 0.2°. Images with voxel resolution of 4.42 µm were reconstructed by the software NRecon (Bruker, Kontich, Belgium). Image analyses were carried out using designated MATLAB codes, the software CT-vox (Bruker, Kontich, Belgium) and the open source-image analysis software ImageJ (Collins, 2007).

Polyvinyl chloride (PVC) tubes, 3 cm long and 1.6 cm in diameter, open at the top and sealed at the bottom, were filled with quarry coarse sand (quartz), with mean grain diameter of $\sim 500$ µm (sand characteristics can be found in Nachshon, 2016). The columns were scanned before and following mechanical compaction, in order to detect the impact of compaction on microscale changes of the sand properties, as a function of depth and distance from the source of compression. Compaction was achieved using a PVC shaft that fits exactly the inner diameter of the column. The shaft was slowly pushed downward to compress the sand samples, using a hand-operated press, achieving a one-dimensional confined compression. The samples were compressed down to a decrease of the total length of the sand sample by 2 mm, corresponding to an increase of $\sim 7 \%$ of the packing bulk density of the samples.

The images from the µCT scans were used to analyze grain and pore sizes at the top 7 mm of the sand samples and at a depth of 9–18 mm. Hereafter, the top and lower levels of the compacted samples will be referred to as "TC" and "LC", respectively, and the non-compacted control will be referred to as "UN". The TC and LC results were used to compare the impact of compaction at the top and the lower layers of the sample. Each µCT scan generates hundreds of images of 2D slices of the sample, with a 4.42 µm distance between adjacent slices. For each scan, five 2D images, out of the hundreds of images, were chosen randomly, processed, and analyzed by a MATLAB code.

Grayscale calculations were based on Otsu's method, which selects the threshold to minimize the interclass variance (Otsu, 1979). Morphological operations were done to clean image noise. Grain counting and grain area calculations were done using the function *regionprops*, with MATLAB –

Image Processing Toolbox. A distance heat map was generated using the Euclidean distance transform, *bwdist* and the function *bwskel*, both from MATLAB – Image Processing Toolbox. Pore sizes were obtained by calculating the average maximal pore distance from the closest grain edge along the pores and throats. A grain distribution map was generated by counting the center of each grain. Size-based segmentation and visualization of the sand grains was done using the "Analyze Particles" function in ImageJ.

## 3.2 Macroscale changes at the soil profile due to compaction

While the µCT experiments described above were used to study the effect of compaction at the pore scale, a transparent 10 cm long and 5 cm in diameter PVC column was used to examine the effect of compaction at the macroscale. The same coarse sand as detailed above was used in this experiment. To allow for visual observation of changes in the compacted sand column, 10 % of the sand (by weight) was colored with a standard red spray paint. The colored sand was thoroughly mixed with the regular sand before packing the column.

As described previously, the sand in the column was compacted by decreasing the total length of the sand sample by 5 mm, using a uniform hammer beating on a circular shaft, with the same diameter as the inner diameter of the PVC column. The bulk density of the sand sample was increased by $\sim 5 \%$ following compaction.

Pictures of the sand column profile were taken before and after compaction by a single-lens reflex camera (Canon – EOS60D, Japan), with an EFS18–200 mm lens (Canon, Japan). Compaction was evaluated by visual analysis of the images that captured the movement of the colored sand grains and measuring the translocation of the same colored grains before and after compaction.

## 3.3 Numerical model

A preliminary analysis was carried out based on simulations using HYDRUS-1D (Šimůnek et al., 2013), in order to estimate the general impact of soil compaction on evaporation, with and without the presence of a precipitated salt crust. Four modeled setups were considered. (i) A homogeneous coarse-texture domain was considered, mimicking natural sandy soil with no compaction. Hereafter, this will be referred to as HC (Homo-Coarse). (ii) A homogeneous coarse-texture domain that underlies a thin layer of a fine texture domain (1 cm) was considered, mimicking compaction of the very top layer of the soil profile. Hereafter, this structure will be referred to as HCC (Homo-Coarse-Compacted). (iii) A layered structure domain was considered, which mimics a compacted soil where the effect of compaction is gradually decreased with depth. This modeled domain is composed of five discrete layers, 2 cm each, where the uppermost layer

had the smallest grain diameter, lowest saturated hydraulic conductivity, and highest air entry pressure. Underlying layers were gradually comprised of bigger particles, higher hydraulic conductivity, and lower air entry pressure. Hereafter, this layered structure with the fine texture matrix at its upper levels will be referred to as FU (Fine-Up). Finally, (iv) a homogeneous domain was considered, which was composed of a homogeneous mixture of the particles that were used to build the discrete layers of the layered structure. By opposition to the homogeneous coarse-texture domain, this structure mimics an uncompacted soil composed of particles with a wide range of particle sizes. Hereafter, this structure will be referred to as HM (Homogeneous-Mix).

Table 1 details the different arrangements of the four modeled structures, with information on their hydraulic properties and particle sizes. The sizes of the particles in the different modeled structures were determined upon real physical sizes of glass beads and coarse-texture sand that were used in column experiments that will be presented in the next section. Saturation water content, $\theta_s$, was determined experimentally by measuring the volume of water needed to saturate the different media, which were packed in a known volume. In order to enable a complete drying of the media, by evaporation, $\theta_r = 0$ was selected, as recently done by Zhou et al. (2021). Nevertheless, the model was tested for $\theta_r$ values in the range of 0–0.07, and for all $\theta_r$ values the simulated results were consistent, with small differences in cumulative evaporation ($<10\%$) and identical trends of water content and pressure head profiles. The van Genuchten parameter $\alpha$ was determined according to Benson et al. (2014) that correlated $\alpha$ to particle diameter. The van Genuchten $n$ parameter is affected by the degree of grain uniformity in the domain (Wang et al., 2017), where high $n$ values indicate high uniformity. Therefore, $n$ was taken as 3 for the uniform layers, as it was the highest $n$ value permitted by HYDRUS, while keeping the relative error in the water mass balance of the entire flow domain at low values on the order of 1 % and below. For the HM domain, $n$ was arbitrarily chosen to be equal to 1.25 as the medium was composed of particles with various sizes. Hydraulic conductivity at saturation, $K_s$ [cm d$^{-1}$], was determined by the Kozeny–Carman equation (Carman, 1937; Kozeny, 1927), as demonstrated by Weisbrod et al. (2013).

The hydraulic properties of a salt layer are unknown, excluding permeability, $k$, which was recently examined and found to be on the order of $4 \times 10^{-12}$ m$^2$, for NaCl (Nachshon and Weisbrod, 2015; Piotrowski et al., 2020). The permeability was used to calculate the saturated hydraulic conductivity of the salt by the relation between $K_s$ and $k$ (Kasenow, 2002):

$$K_s = \frac{k \cdot \rho \cdot g}{\mu},\tag{1}$$

where $\mu$ [kg m$^{-1}$ s$^{-1}$] TS2 is the dynamic viscosity, $\rho$ [kg m$^{-3}$] is the liquid density, and $g$ [m s$^{-2}$] is gravity acceleration. For water, $\rho = 1000$ kg m$^{-3}$, and $\mu \sim$ 0.0009 kg m$^{-1}$ s$^{-1}$ TS3 (at 25 °C). $g = 9.8$ m s$^{-2}$, and for the NaCl permeability of $4 \times 10^{-12}$ m$^2$, $K_s$ is equal to $4.3 \times 10^{-5}$ m s$^{-1} = 376.0$ cm d$^{-1}$. Since no further information is available about the salt hydraulic properties, the van Genuchten parameters of the salt layer were taken to be equal to loamy-sand soil, from the HYDRUS-1D library, due to the similar hydraulic conductivity that this soil (350.2 cm d$^{-1}$) has to the salt layer. What is important to emphasize is that the model examined only the physical impact that a salt crust has on water flow process during S1, and it did not account for the chemical aspects of high salinity and associated changes of the solution osmotic potential, surface tension, or viscosity. Salt crust hydraulic properties are also depicted in Table 1.

The modeled domains had a depth of 10 cm, and the upper boundary condition was set as atmospheric boundary, with potential evaporation of 0.65 cm d$^{-1}$ (based on the data obtained in the laboratory glass-bead evaporation experiment, which will be detailed below). The lower boundary was set as zero flux, and initial condition was set as full saturation throughout the entire column. Since HYDRUS solves the Richards equation, its results are valid only during S1 where evaporation occurs at the soil surface and there is a hydraulic continuity along the soil profile. Therefore, simulations were ceased once S1 was ended and the transition to S2 had begun.

The simulations were used to observe changes of the soil profile wetness and to compute length of S1 and the impact of compaction and salt precipitation on its length. The salt crust was simulated by adding a 2 mm layer of the crust on top of the modeled domains (this thickness is similar to the observed one, corresponding to the depositing salt layers during the experiments presented below). This layer was added after 2 d of evaporation, as it is experimentally known that the appearance of the salt crust is not instantaneous with the onset of evaporation.

The upper boundary condition of the simulated salt layer was as defined for the salt-free setup, with atmospheric potential evaporation of 0.65 cm d$^{-1}$. Initial pressure head of $-1000$ cm was defined for the added salt layer, assuming it is dryer than the underlying soil. Model sensitivity to the initial pressure head of the salt crust was low, as it was tested for various levels in the range of $-1000$ to $-100$ cm, and simulation results were identical, as after one time step the pressure head of the salt crust and resultant water content were adjusted with respect to the wetness of the underlying soil. What is important to emphasize is that the model of the FU structure was also tested for a more moderate change of the soil hydraulic properties, where the hydraulic properties of the five layers (Table 1) were interpolated and evenly distributed over 30 layers (3 mm each). Results of the five layers and the 30 layers' structures were similar; hence, hereafter only the results of the five layers will be discussed, as they correspond to the experimental setup.

**Table 1.** Hydraulic parameters of the modeled setups. Symbols $\theta_s$, $\alpha$, $K_s$, and $d$ stand for water content at saturation ($cm^3\,cm^{-3}$), the $\alpha$ van Genuchten parameter ($cm^{-1}$), hydraulic conductivity at saturation ($cm\,d^{-1}$), and range of particle diameters (mm), respectively. Where not mentioned, the $n$ van Genuchten parameter is equal to 3 (unitless). For all simulations, the tortuosity parameter in the conductivity function was taken to be equal to 0.5 (–).

| Depth (mm) | Homo-Coarse (HC) | Homo-Coarse-Compacted (HCC) | Homogeneous-Mix (HM) | Fine-Up (FU) |
|---|---|---|---|---|
| | | | Setup | |
| 0–10 | | $\theta_s = 0.39$ $\alpha = 0.01$ $K_s = 233$ $d = 0.049\text{–}0.053$ | | $\theta_s = 0.39$ $\alpha = 0.01$ |
| 10–20 | | | | $K_s = 233$ $d = 0.049\text{–}0.053$ |
| 20–40 | | | $\theta_s = 0.28$ $\alpha = 0.05$ $n = 1.25$ | $\theta_s = 0.36$ $\alpha = 0.02$ $K_s = 578$ $d = 0.090\text{–}0.106$ |
| 40–60 | $\theta_s = 0.29$ $\alpha = 0.07$ $K_s = 5655$ $d = 0.4\text{–}0.5$ | $\theta_s = 0.29$ $\alpha = 0.07$ $K_s = 5655$ $d = 0.4\text{–}0.5$ | $K_s = 3537$ $d = 0.09\text{–}1.3$ | $\theta_s = 0.38$ $\alpha = 0.03$ $K_s = 2922$ $d = 0.18\text{–}0.212$ |
| 60–80 | | | | $\theta_s = 0.29$ $\alpha = 0.07$ $K_s = 5655$ $d = 0.4\text{–}0.5$ |
| 80–100 | | | | $\theta_s = 0.41$ $\alpha = 0.15$ $K_s = 14\,613$ $d = 1.0\text{–}1.3$ |
| | | Salt crust | | |
| 2 mm above soil surface | $\theta_s = 0.41$, $\alpha = 0.124$, $n = 2.28$, $K_s = 350$ | | | |

## 3.4 Impact of compaction on evaporation and salt precipitation

Two sets of column-evaporation experiments were conducted: (i) columns filled with glass beads at varied arrangements, mimicking different conditions of non-compaction and compaction scenarios, and (ii) columns filled with coarse sand under compaction and non-compaction conditions. The glass bead experiments aimed to test the research hypothesis under synthetic and controlled conditions, whereas the coarse sand column experiments aimed to better correlate the synthetic structures results to a more natural setup, with a deeper soil profile.

The glass-bead evaporation experiments were conducted on rectangular glass columns: 10 cm height, 5 cm width, and 2.5 cm aperture. Corresponding to the numerical models, the glass-bead experimental setups were identical to the HM and FU structures, as detailed in Table 1. In addition, in this set of experiments, a setup mimicking a tilled soil was also examined, and for this purpose, the tilled setup was constructed in a reverse order of the FU setup, with the largest glass beads being located at the top of the profile and the smallest beads at the bottom. Hereafter, this structure will be referred to as CU (Coarse-Up). The three different setups were saturated with distilled (DI) water or with a 10 % (by weight) NaCl solution. All the evaporation experiments were carried out in two replicates. The packed columns were positioned on high-resolution electronic scales ($\pm 0.01$ g, Adam; Shekel, Israel) in order to record mass changes, thus monitoring the cumulative water loss to evaporation. Small fans (Y.S. Tech, DC brushless fan, FD128020HB, DC 12 V, 0.15 V) were in-

stalled $\sim 3$ cm above the upper soil surfaces of the samples, pulling air upward. Along the process of evaporation, photos of the column profiles were taken with a camera (UEye, Germany) at a rate of six pictures per minute. Total duration of evaporation for each setup was about 12.5 d ($\sim 303$ h). This set of column experiments and the corresponding numerical models detailed above were carried out for columns shorter than the capillary lengths of the porous media under interest. Consequently, the resulting duration of S1 is affected by this physical constraint (Assouline et al., 2014). In order to expend and to validate the findings of the small column experiments to more realistic conditions, the coarse sand column experiments were conducted as detailed below.

The coarse sand experiments were conducted in circular columns: 92 cm long and 4.1 cm in diameter. The columns were filled with the same coarse sand used for the CT scan experiments. Soil compaction, which lowered the soil surface by 4 cm, was achieved as detailed in Sect. 3.2. The columns were saturated from the bottom, through a designated valve, by DI water or 10 % NaCl solution. After saturation, the columns were placed in the laboratory under the small fans, as was done for the glass-bead columns. Every few hours the columns were weighed on a 0.2 g accuracy scale (Snowrex NHV-6, Sam Hing Scales Factory Limited, Kowloon, Hong Kong). The experiment lasted for 250 h, with two repetitions for the compacted and uncompacted DI setups. Even though the dimensions are not exactly similar, these setups correspond to the modeled HC and HCC structures.

## 4    Results and discussion

Experimental results are organized and presented first for the microscale and then for the macroscale, considering the physical changes that the sand underwent due to compaction. Following that, the results representing the impact of compaction on the combined processes of evaporation and salt precipitation will be discussed.

### 4.1    Microscale effects of compaction

The impact of compaction on changes in grain and pore geometry, size, and distribution, at the microscale, was examined by producing 3D and 2D images (slices) of the sand domain using the µCT (Fig. 2). The 2D slices were randomly selected along the vertical axis of the columns and used to quantitatively analyze the different physical properties of pores and grains of the compacted and uncompacted samples. Figure 3 presents the analysis process which was done for the 2D slices, in order to observe the changes in pore and sand grain properties following compaction at the top layer of the sample and at its bottom. Figure 3a shows representative images from (i) an uncompacted sample (UN) at a depth of 0–7 mm (Fig. 3a′), (ii) the lower part of the compacted sample (LC; depth of 9–18 mm) (Fig. 3a″), and

(iii) the top part of the compacted sample (TC; depth of 0–7 mm) (Fig. 3a‴). In the UN, as well as in the LC domain, the sand grains are relatively round and uniform in size. By comparison, in TC, there are areas with a high proportion of relatively small and more angular grains, a result of the grains breakage in specific locations (marked by the yellow contours in Fig. 3a‴ and also depicted in Fig. 2). Naturally, these changes in grain sizes also affect pore sizes and their spatial distribution, as visually observed in Fig. 2, and depicted by the pore openings heat map (Fig. 3b).

In Fig. 3c, we demonstrate the changes in number of grains for a given area in the TC sample and the spatial distribution of these changes in comparison to the UN and LC data. For this purpose, each 2D scan, of any state and depth, was divided into a matrix of rectangles: 1.06 mm by 0.73 mm each. In each rectangle, the number of sand grain centers was counted, and the rectangle was colored in accordance with the number of grain centers. In the presented images, the main colors for the LC and UN cases are blue and green, indicating about 3–4 grain centers per rectangle, with low variation in colors. However, for the TC case, there is a high variation in the color of the rectangles, with a relatively high number of yellow and red rectangles (>6 grain centers) adjacent to green and blue rectangles.

The five randomly selected images of "UN", "TC", and "LC" states were averaged and analyzed, as demonstrated in Fig. 3, to provide a corresponding quantitative analysis of the number of grains per unit area, grain size (2D area), and pore openings (distance between adjacent grains) (Fig. 4). For simplicity, all of these values were normalized with respect to those corresponding to the UN state. In agreement with the visual observations, minor differences were measured with respect to the number of grains between the UN and LC states. However, a significant difference was measured with the TC samples, where the total number of grains per unit area was $\sim 50$ % higher for TC compared to UN and LC (Fig. 4a). Moreover, with respect to changes in grain sizes, there is no significant difference between UN and LC, but for the TC layer, the average size of the grains was $\sim 35$ % lower compared to UN and LC cases. The same trend was also measured with respect to pore openings, as the pore average opening of the TC was lower by $\sim 10$ % compared to the two other cases.

The analysis of the grain counting within the rectangles (Fig. 3c) was also conducted for the five randomly selected images. Analysis of each image was used to generate a histogram describing how many rectangles were contained in the different numbers of grains (Fig. 4b). For the UN and LC cases, grain density was lower compared to the TC setup, where the former had on average 2–2.5 grains per rectangle, whereas the latter had 4 grains per rectangle. Moreover, for the TC layers, in comparison with the LC and UN cases, the histogram shifts to the right, indicating a higher number of rectangles that contain 4 grains or more.

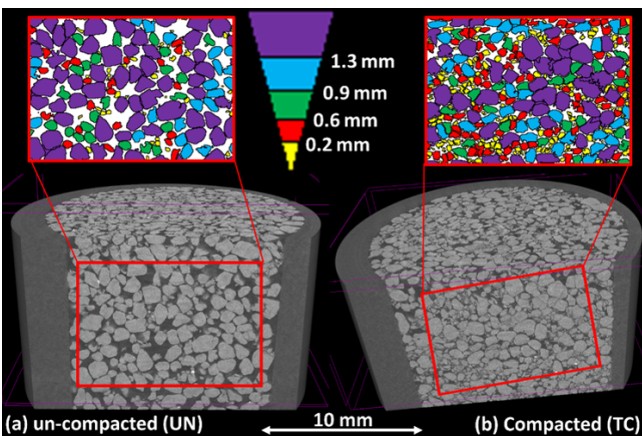

**Figure 2.** A 3D visualization of the top $\sim 10$ mm of uncompacted **(a)** and compacted **(b)** sand samples. Red rectangles exemplify the cropping of the 2D slices from the 3D structure. The 2D colored slices present, visually, the different grain size distributions between the compacted and uncompacted samples. Grains sizes are given with respect to the color bar. In addition, it is seen in **(b)** that compaction reduced pores sizes and increased grains angularity, due to breakage of the grains.

The presented image analyses, at the microscale, indicate the major impact that compaction has on the physical properties of the sand, close to the source of compaction (TC). It seems that compaction resulted in breakage due to friction of sand grains, leading to an increase in the grain number and their angularity and a decrease in their mean size. In the deeper layer of the sand column (LC), the grains were practically not affected by the compaction and were similar to the control (UN) with respect to pore and grain sizes, shape, and spatial distribution. Moreover, it was shown that the compacted areas in the top layer were heterogeneously distributed (Figs. 3c, 4b), in agreement with the concept of preferential propagation of the stress along the "shear bands" (Naveed et al., 2016; Nawaz et al., 2013).

### 4.2 Macroscale effects of compaction

At the macroscale, compaction effects were quantified by following the translocations of the colored sand grains in the 10 cm long transparent column. The translocation of the sand grains, $\Delta L$ (mm), was calculated by measuring the distance of selected grains from the column's bottom at the initial state ($L_0$) and following compression ($L_C$), according to the following:

$$\Delta L = L_C - L_0. \tag{2}$$

Figure 5 presents $\Delta L$ along the soil profile. Maximal translocation was observed within the upper layer of the soil profile, and it linearly decreases with depth, in agreement with the results reported by Schlüter and Vogel (2016). However, it is important to remember that translocation of the grains at each depth is the sum of all compaction processes that occurred below the point of interest and that it does not necessarily indicate the degree of compaction (change of bulk density) at this point. In order to estimate the effect of compaction on the bulk density along the soil profile, we estimated the changes in distance between adjacent grains, $\Delta D$ (mm), in a similar way that was done for $\Delta L$:

$$\Delta D = D_C - D_0, \tag{3}$$

where $D_0$ and $D_C$ are the measured distances between any adjacent selected grains before and following compaction, respectively. Consequently, a negative $\Delta D$ value indicates compaction and increase in bulk density and vice versa.

Measurements of $\Delta D$ indicate that compaction was not uniform along the sand profile (Fig. 5), showing that certain depths were more severely compacted. At depths of 6, 17, 36, and 60 mm, $\Delta D$ values were positive, indicating reduced bulk density at these specific locations. Maximal compaction of $\Delta D = -1.3$ mm was measured at a depth of 32 mm, followed by $\Delta D = -1.1$ mm at a depth of 8 mm. Lower levels of the column were less compacted, excluding depths of 68 and 82 mm, where $\Delta D$ reached values of $-1.0$ and $-0.8$ mm, respectively.

This analysis further emphasizes the heterogeneous nature of soil compaction and the shear band effect, as different locations along the profile were more compacted compared to others. These differences are more notable at the macroscale compared to the microscale observations from the CT experiments. Nevertheless, it is evident that most of the profile underwent compaction, as most of the $\Delta D$ values are negative, and that maximal compaction was at the top $\sim 30$ mm of the sample.

As seen from the microscale and macroscale experiments, compaction induces the formation of a non-uniform soil profile, with smaller pores, smaller grains, and higher bulk density at the top levels of the soil profile compared to the lower part of the profile. This structure is opposed to typical natural conditions, where the lower soil levels are those with the higher bulk density (Campbell, 1994; Hernanz et al., 2000). Consequently, important hydrological processes such as infiltration and evaporation may be altered due to compaction. These aspects will be discussed in the following sections.

### 4.3 Numerical model

The HYDRUS-1D model results showed similar trends to those proposed in the conceptual model and in the research hypothesis, including the unique pattern of drying from bottom to top, for the simulated compacted structures, with the fine grains at the top (HCC, FU). Figure 6 presents the length of S1 for each one of the simulated structures, for conditions of with and without a salt crust. It is seen that for salt-free conditions, the compacted structures had the longest S1 du-

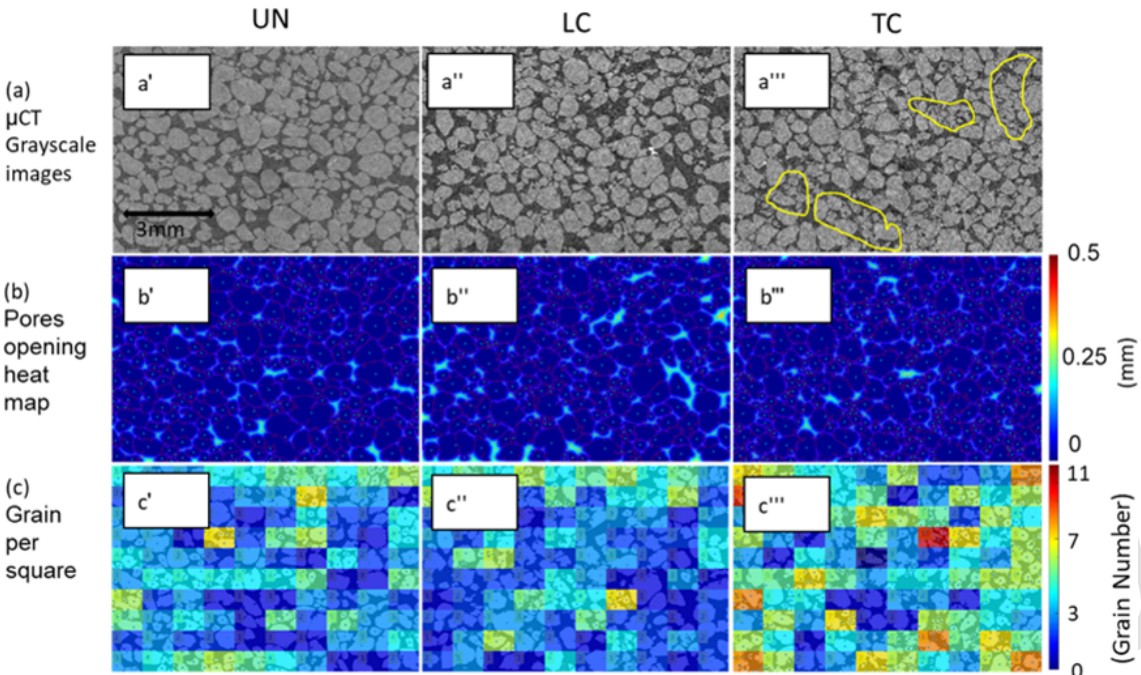

**Figure 3. (a)** μCT grayscale cross-section scans; **(b)** pore openings heat maps; and **(c)** grain number spatial distribution maps for the uncompacted (′), low (″), and top (‴) levels of compacted soil samples, respectively. Yellow contours in (**a**‴) indicate areas with high levels of grain breakage. In **(b)**, the blue dots represent the grain centers, and the color bar indicates the distance within the pores from the nearest grain. In **(c)**, the rectangles are colored in proportion to the number of grains in each one of them.

rations, with 136 h for the FU structure and 108 h for the HCC setup. This was followed by the HC setup (96 h) and the HM (72 h). For all cases, the addition of the salt crust resulted in shortening of S1, yet the compacted structures had the longest S1 durations, also for the saline conditions, whereas, for the homogeneous structures, S1 durations were the shortest (Fig. 6).

Figure 7 displays water content distribution and effective hydraulic conductivity values along the modeled domains, 48 h after evaporation onset. This time is also when the simulated salt crust was added for the saline setups. These profiles give the physical explanation for the extended S1 duration of the compacted structures for both with and without the addition of the salt crust. It is seen for the FU and HCC setups that the fine-grain-texture domain on top of the coarse-texture domain results in drying of the matrix from top to bottom, due to the stronger capillary suction of the finer pores. Consequently, higher water content levels are maintained close to the soil surface of the compacted setups, where evaporation is maximal, while the underlying coarse-texture regions act as a water reservoir to replenish evaporation (Fig. 7a).

For all the considered cases, the addition of the salt crust resulted in reduction of S1, after some time, indicating breaking of the hydraulic continuity between the underlying wet matrix and the salt crust. Also in this perspective, the elevated water content levels of the upper regions of the compacted structures resulted in higher levels of hydraulic con-

ductivities (Fig. 7b), which could support a capillary flow of liquid water from the deeper parts of the soil matrix to the salt crust and by that maintain a longer S1 compared to the non-compacted setups.

Model results point on the high impact that the hydraulic properties of the simulated domain have on the evaporation dynamics. For example, notable differences were observed between the two homogeneous and non-compacted setups with respect to S1 length, wetness profiles, and impact of the salt crust on evaporation. These differences were a result of the different $n$ values of the two domains, as most other hydraulic parameters were relatively similar. In addition, when the $n$ value of the HM was elevated to be equal to the $n$ value of HC, the simulated results were similar. Future studies should further explore these disparities and the sensitivities of the modeled systems to the different hydraulic parameters. However, for the purpose of this study, simulation results strengthen the conceptual model and research hypotheses (Fig. 1), which were further explored by the following column experiments.

## 4.4 Evaporation and salt precipitation in layered glass-bead domain

As previously mentioned, glass beads were used to fill the rectangular columns, as detailed in Table 1, to represent the compacted conditions (FU) and in a reverse order (CU) to

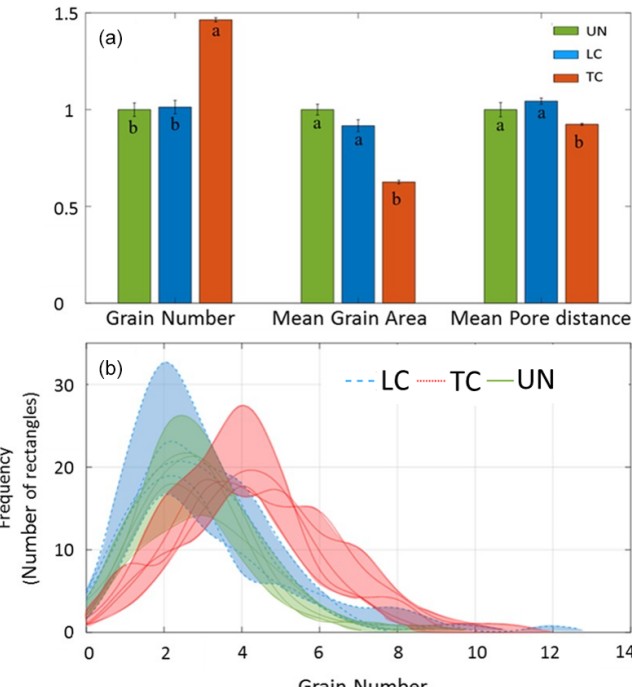

**Figure 4. (a)** Average grain number, average grain area, and average pore distance of the uncompacted soil sample (UN), low (LC), and top (TC) levels of compacted soil samples. Measured values were normalized with respect to UN. **(b)** Histogram represents grain number distribution of the uncompacted soil samples and low and top levels of the compacted soil samples. In **(b)**, each line is the histogram of a single slice, and the colored areas represent the range of the five measured histograms for each state.

mimic loose soil, e.g., tilled soil CE2. The HM setup was also experimentally tested, representing a non-compacted structure with a wide range of particle size distribution. The evaporation process during the column experiments of the three glass-bead configurations saturated with the saline solution is represented by a set of pictures in Fig. 8. It is possible to follow the movement of the drying front for the homogeneous configuration (HM) and the compacted (FU) structure, which coincides with the numerical model results. Water flow and drying processes are also observed for the tilled (CU) setup.

For the HM structure, the evaporation front receded from top to bottom, as typically seen in evaporation of homogeneous porous media. A notable efflorescent salt crust is observed in the image of 54 h (Fig. 8a), yet it is important to emphasize that salt precipitation started at about 15 h following evaporation onset.

For the FU structure, the soil surface remained moist for the entire duration of the experiment, while the drying front progressed upward from bottom to top (Fig. 8b). The unique drying pattern of the FU structure, which mimics compacted soil, is a result of the hydraulic properties of the top layer that had the highest capillary suction along the soil profile. This structure results in a continuous upward flow of the so-

lution from the coarser layers at the bottom that have a lower capillary suction. Replacement of water by air in the coarse-texture layers points to air invasion into the matrix, from the soil surface, via air conduits which are likely to develop in the relatively larger pores of the fine-texture layer. A similar behavior was reported by Assouline and Narkis (2019) for DI water, where a detailed explanation of the impact of layered structure on evaporation is given. For the FU setup, under saline conditions, evaporation resulted in salt precipitation, at the soil surface, as observed herein in the image taken after 54 h of evaporation (Fig. 8b). As detailed for the HM setup, salt precipitation initiated after about 15 h.

For the CU structure, air penetration into the coarse upper layer was observed after 14 h of evaporation, and a slow recession of the evaporation front downward was observed over time (Fig. 8c). For the CU case, salt precipitation was minor, due to the receding evaporation front, with no formation of a salt crust on the surface or inside the medium.

Measurements and recording of changes in column masses during the experiment enabled us to compute average cumulative evaporation of the different setups, as presented in Fig. 9. Maximal values of standard deviation, for each setup, are detailed in Table 2. Transition from S1 to S2, which is the time at which the initial high and constant evaporation rate started to decrease, is marked on each cumulative evaporation curve in Fig. 9. The transition times were determined by identifying the divergence of the curves from their tangents during initial stages of evaporation (see example for FU setup in Fig. 9). The slope of each tangent line describes the initial (S1) evaporation rate of each curve.

For the HM, the duration of S1 with DI water was about 46 h, with a cumulative evaporation of $\sim 14$ mm. Total evaporation after 300 h for the HM, with DI water, was 23 mm. The relatively long S1 duration and high cumulative evaporation for HM resulted in the formation of a notable efflorescent salt crust (Fig. 8a), with a thickness of about 3.5 mm as estimated from the images. The saline conditions reduced the duration of S1 by more than 70 %, and cumulative evaporation at the transition from S1 to S2 was lowered by more than 80 % compared to evaporation from initially DI-saturated HM columns.

In agreement with the observed drying pattern (Fig. 8b), it was shown that FU S1 was the longest compared to all other setups (Fig. 9). S1 duration for the FU structure was of 66 and 62 h for the DI and saline conditions, respectively. Cumulative evaporation was also high for the FU setup, with 11 and $\sim 18$ mm at the end of S1 for the saline solution and DI conditions, respectively, and total cumulative evaporation after 300 h of 26 mm for the saline solution and 28 mm for the DI water. In comparison to HM with DI water, the cumulative evaporation of the FU, after 300 h, was 13 % higher (Fig. 9). The long duration of S1 for FU, the persistence of the evaporation front at the surface of the column, and the corresponding high cumulative evaporation (Fig. 9) led to the

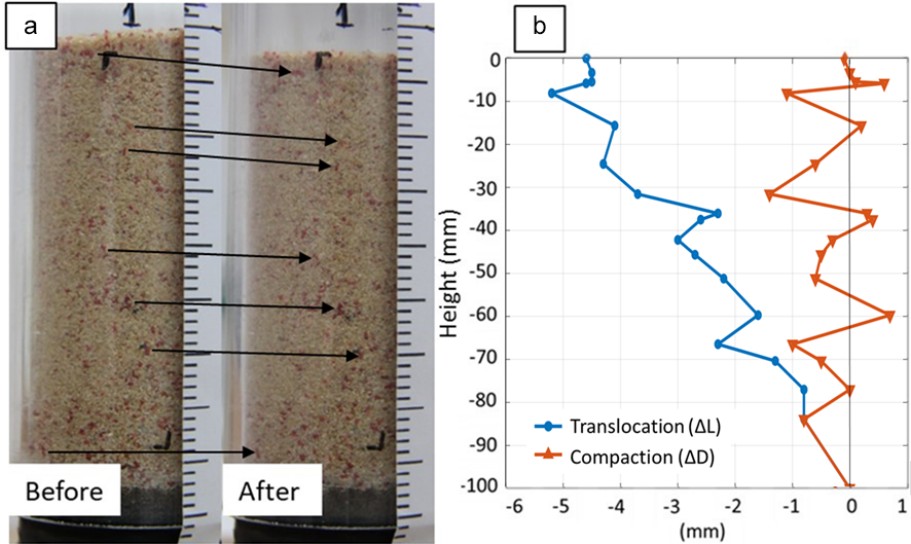

**Figure 5. (a)** Photos of the examined sand column "Before" and "After" compaction. Black arrows exemplify the vertical transition, due to compaction, of selected colored grains; **(b)** measured changes in grain translocation and compaction along the sand column.

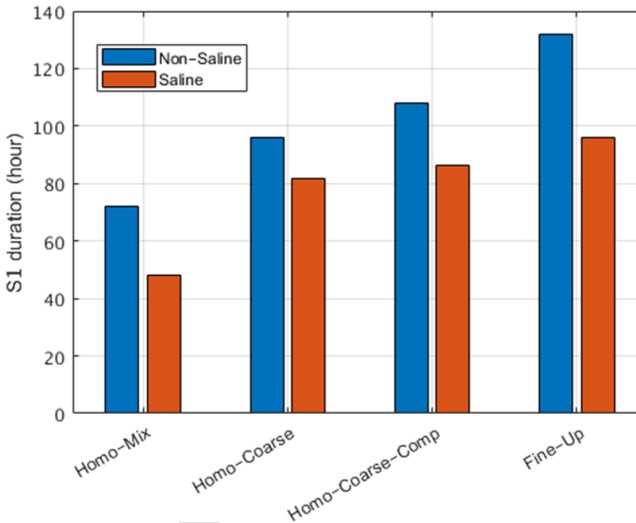

**Figure 6.** Duration of S1 for the four modeled setups, with and without the formation of the salt crust.

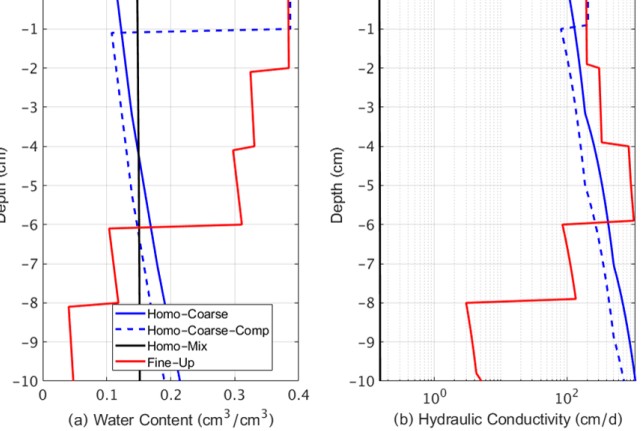

**Figure 7.** Simulation results of water content **(a)** and hydraulic conductivity **(b)** along the four modeled setups 48 h after evaporation onset.

precipitation of a notable efflorescent salt crust (Fig. 8b) with a thickness of about 6 mm as estimated from the images.

While saline conditions for the HM setup led to reduction of more than 50 % in total evaporation and major changes in duration and cumulative evaporation of S1, for the FU setup the impact of salinity was much less prominent. For FU, the salinity reduced cumulative evaporation and the duration of S1 by less than 10 %, and cumulative evaporation at the transition from S1 to S2 was reduced by less than 40 % (Fig. 9).

Unlike for the HM setup, the evaporation from the CU column showed a transition from S1 to S2 after $\sim$ 19 h of evaporation and a cumulative evaporation of $\sim$ 3.5 and $\sim$ 4 mm

(Fig. 9) for the saline solution and DI water, respectively. These results and the relatively quick transition into S2 coincide with the receding of the drying front downward as seen in Fig. 8c. During S2, evaporation was minimal, due to the low rate of vapor diffusion through the dry coarse porous medium at the surface, and total cumulative evaporation after 300 h was 10 mm for both saline and DI conditions, which is less than half the cumulative evaporation of the HM setup and $\sim$ 61 % lower than the FU. As previously mentioned, for the CU saline conditions no salt crust was observed because of the receding evaporation front that did not support the processes of salinity buildup at the soil surface. The absence of the salt crust at the surface of the CU column explains the

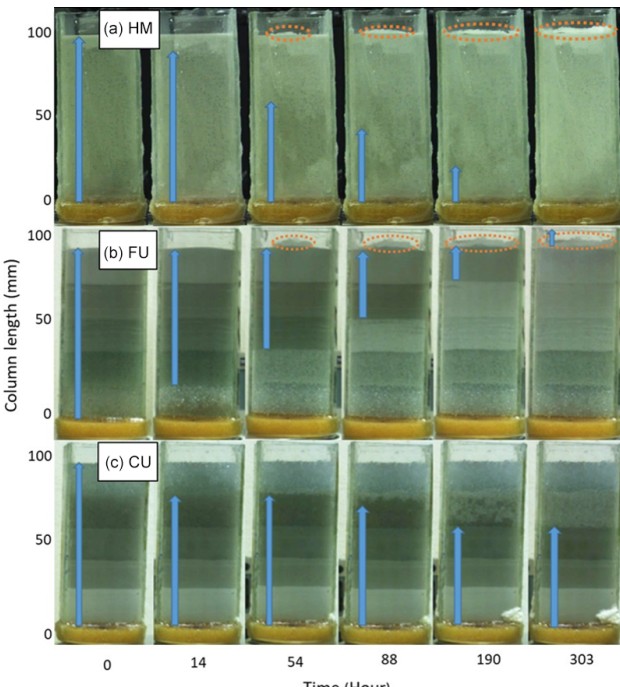

**Figure 8.** Evaporation and salt precipitation patterns for **(a)** HM – homogeneous mixture of the glass beads, **(b)** FU – with the fine glass beads at the upper levels, and **(c)** CU – with the coarse glass beads at the upper levels. Numbers at the bottom indicate time length of evaporation. Top and bottom of the blue arrows indicate the upper and lower boundaries of the saturated zones, respectively. The orange ovals mark locations of salt precipitation.

**Table 2.** Maximal values of standard deviation for cumulative evaporation measurements.

|      | DI water [mm] | Saline solution [mm] |
|------|---------------|----------------------|
| HM   | 5.833         | 3.312                |
| FU   | 5.928         | 4.957                |
| CU   | 1.297         | 0.676                |

observed negligible difference in evaporation between the saline and DI water setups (Fig. 9).

## 4.5 Differences in salinity impact on evaporation

As shown above, the three different setups (HM, FU, and CU) responded differently for the saline conditions, with the greatest impact observed for HM, followed by the FU, and CU that presented minimal changes. Figure 10 presents the relative change in cumulative evaporation for the different setups over time. After $\sim 5$ h of evaporation, all setups presented a reduction of about 30 % in cumulative evaporation compared to the DI conditions. This reduction in evaporation may be a result of increased pore water NaCl concentration near the evaporation front at the surface of the columns,

which results in reduction of the solution osmotic potential and vapor pressure. The 30 % reduction coincides with the fact that vapor pressure of a saturated NaCl solution at 25 °C is equal to 2.401 kPa, which is $\sim 25$ % lower than the vapor pressure of pure water that is equal to 3.169 kPa (Lide, 2007). However, more interestingly, after these first 5 h, the relative impact of salinity on evaporation started to vary significantly, depending on the soil structural configurations.

The HM setup introduced a reduction in evaporation that was much greater than 25 %, on the order of 60 %, throughout most of the evaporation process, with a maximal reduction of $\sim 70$ % after about 50 h. Total reduction in cumulative evaporation at the end of the experiment was around 50 % (Fig. 10). For the FU setup, the reduction in cumulative evaporation was maintained at 30 %–35 % for about 100 h, which is approximately 35 h longer than S1 duration of the DI setup ($\sim 65$ h). After $\sim 100$ h, the difference between the saline and DI setups for FU was gradually moderated, along S2, and by the end of the experiment, total cumulative evaporation of the saline FU setup was only 10 % lower compared to the DI state (Fig. 10). For the CU setup, after the initial reduction of $\sim 30$ % at the first 5 h of evaporation, the difference between the saline and DI conditions decreased to very low values, and after 150 h of evaporation, no differences were observed (Fig. 10).

For both HM and FU setups, the greatest difference in evaporation between the DI and saline conditions was observed during the time when the DI columns were at S1 and the saline solution configurations moved into S2. The large difference between the DI and saline conditions for HM during this time, on the order of 70 %, indicates that the reduction of the solution vapor pressure is not the only mechanism that reduces evaporation. For the non-saline condition, the S1–S2 transition occurred after $\sim 50$ h, for the HM setup. However, for the saline condition, S2 started after $\sim 10$ h of evaporation only, with a minor cumulative evaporation on the order of 2 mm.

From the HM-DI setup, it is understood that, at that time, the saline domain is moist enough to supply water to the upper-atmosphere–domain interface and that S1 should be sustained. Therefore, it is concluded that the transition into S2 after $\sim 10$ h is likely a result of increased osmotic potential of the solution, salt precipitation, and the development of the efflorescent salt crust on top of the HM domain. The precipitated salt crust acts as a mulching layer that results in hydraulic discontinuity between the saturated domain and the atmosphere. Even though the matrix under the salt crust is moist enough to support S1 under salt-free conditions, it is not wet enough to support liquid water flow into the salt crust; therefore, vapor diffusion through the salt dictates the evaporation rate. This is in agreement with observations from previous studies (Gran et al., 2011; Nachshon and Weisbrod, 2015) and the numerical simulation.

For the FU setup, the fact that the differences in duration of S1 between the DI and saline conditions were minor (Fig. 9)

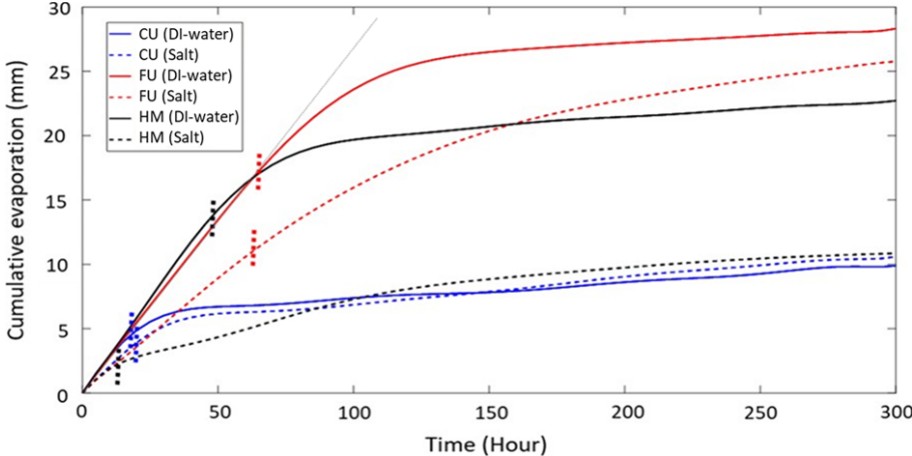

**Figure 9.** Cumulative evaporation for the Coarse-Up (CU), Fine-Up (FU), and the Homogeneous-Mixed (HM) structures for conditions of DI water (solid lines) and saline solution (dashed lines). Vertical dotted lines indicate the transition from S1 to S2. The thin gray line is an example of a tangent line used to identify the S1–S2 transition.

and the reduction in cumulative evaporation during S1 is on the order of 30 %–35 % suggest that the main mechanism that reduced evaporation was the reduction of the solution vapor pressure. The negligible impact of precipitated salt crust, for 5 the FU setup, suggests that in this case the crust was hydraulically connected to the underlying media and that liquid water was flowing towards the surface of the salt crust, where the evaporation front was located. This hydraulic continuity suggests that the unique structure of the FU state, which mimics 10 compacted soils, enables water from the lower layers of the drying profile to flow upward into and through the salt crust. It is suggested here that the hydraulic continuity between the precipitated salt crust and the underlying domain was possible for FU and not for HM, due to the unique FU structure 15 that keeps the upper layer of the domain wet, as also demonstrated by the numerical model (Fig. 7).

For the CU setup, it is believed that during S1 the increase of the NaCl solution concentration at the evaporation front led to the observed reduction in evaporation on the order of 20 30 %. This is in agreement with the vapor pressure reduction of a saturated NaCl solution. However, during S2 the differences between the DI and saline conditions decreased as vapor diffusivity, through the porous domain, became the factor controlling evaporation for both cases: with and with- 25 out a salt crust.

### 4.6 Sand column experiments

The glass bead experiments and the numerical model supported the research hypothesis. However, both consisted of a layered structure, assuming it is a reasonable approxima- 30 tion for compacted conditions. The advantage of using the layered structure (FU), with the fine-texture media overlying coarser texture domains, is the simplicity of constructing the domain under controlled, accurate, and reproducible

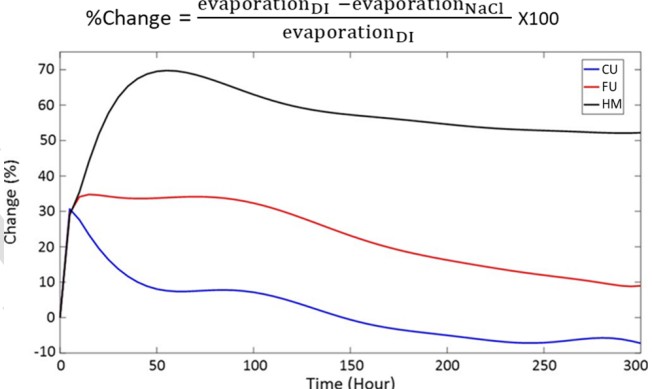

$$\%\text{Change} = \frac{\text{evaporation}_{\text{DI}} - \text{evaporation}_{\text{NaCl}}}{\text{evaporation}_{\text{DI}}} \times 100$$

**Figure 10.** Relative difference in cumulative evaporation between saline and non-saline soil water solution over time, for the tilled (CU), Homogeneous-Mixed (HM), and compacted (FU) setups.

conditions. However, in reality, soil compaction will form a more complex structure, as also shown here by the mi- 35 croscale and macroscale compaction experiments. Moreover, while the HM setup represents a non-compacted homogeneous domain with a wide range of particle size distribution, it fails to be an accurate non-compaction counterpart for the FU setup. As detailed in Sect. 4.1 and 4.2, compaction af- 40 fects mainly the upper levels of the soil profile, while the deeper parts of the profile maintain their initial physical and hydrological properties. Therefore, the initial state of the FU setup, prior to compaction, should be composed of a homogeneous domain with a narrow range of particle size distribu- 45 tion and with a texture similar to the lower levels of the FU profile. Therefore, in order to associate the findings of the glass bead experiments and simulations with real-life conditions, and for more realistic length scales, the experiments

considering compacted and uncompacted sand columns, with $\sim 1$ m length, were conducted.

In agreement with the numerical model simulations and the glass bead experiments, it is seen that the compacted sand, with no salt, had the highest cumulative evaporation with total evaporation of $20.0 \pm 0.23$ mm (Fig. 11). For the uncompacted sand, maximal cumulative evaporation was equal to $16.9 \pm 3.11$ mm. With respect to the S1–S2 transition, in agreement with the layered structure results, it is seen that highest evaporation rate is measured for the DI compacted sand, with a notable reduction in evaporation rate observed after $\sim 48$ h of evaporation and cumulative evaporation of $\sim 8$ mm (Fig. 11). For the uncompacted DI sand, transition into S2 with a notable reduction in evaporation rate was observed after $\sim 25$ h of evaporation, with cumulative evaporation of about 4.5 mm (Fig. 11).

For saline conditions, where salt precipitation was observed for both compacted and non-compacted setups, after S1, the compacted sand also displayed higher cumulative evaporation compared to the uncompacted state, with total cumulative evaporation of about 16.5 and 14.5 mm for the compacted and uncompacted samples, respectively (Fig. 11). S1 duration of the compacted saline setup was approximately twice as long as the non-compacted saline domain ($\sim 40$ vs. 20 h, respectively). However, cumulative evaporation during S1 was only $\sim 40$ % higher for the compacted setup ($\sim 4.8$ mm) compared to the non-compacted domain ($\sim 3.5$ mm), indicating a lower evaporation rate during S1 for the saline-compacted setup compared to all other setups. Future studies should clarify this disparity, yet it could be related to preferential water flows that may be developed in compacted soils (Zhang et al., 2018; Shein et al., 2003; Nagy et al., 2018), due to heterogeneous changes of the compacted soil texture and structure along the stress chains (Nawaz et al., 2013; Naveed et al., 2016). Consequently, evaporation may be concentrated in specific locations at the soil surface and not homogeneously distributed as common for homogeneous domains. This may lead to elevated salt concentration in the pore water in the locations of concentrated evaporation, as demonstrated by Shokri-Kuehni et al. (2020). The resulting increased salt concentration increases the osmotic pressure and reduces vapor pressure of the solution, which reduces evaporation. It is likely that in the glass-bead experiments this phenomenon was not observed since the glass beads layer were homogeneously packed.

Regardless of the reduction of evaporation rate during S1 for the compacted saline setup, which should be further explored, this set of sand column experiments further strengthens the conceptual model and research hypothesis, as being presented in Fig. 1, and indicates that the research findings are valid even though the soil profiles were relatively shallow.

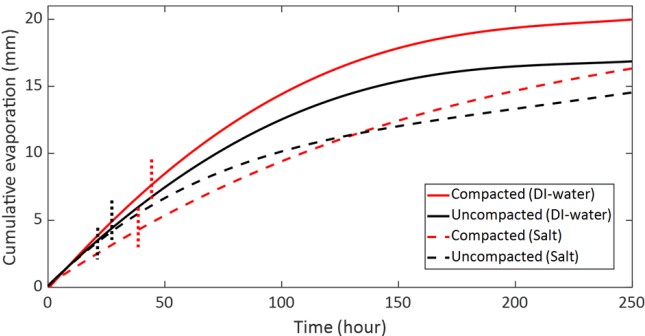

**Figure 11.** Cumulative evaporation for the compacted and uncompacted sand samples for conditions of DI water (solid lines) and saline solution (dashed lines). Vertical dotted lines indicate the transition from S1 to S2.

## 5 Summary and conclusions

This study investigates the effect of compaction on sand physical properties at the microscale and macroscale, as well as its impact on evaporation combined with salt precipitation processes. Microscale properties such as the geometrical pore parameters were studied using X-ray computed microtomography (μCT) techniques by scanning sand samples before and after compaction. Compaction resulted in breakage of sand grains, reduced grain sizes, and pore average opening, mainly close to the source of compression. The spatial distribution of grain number, for the top levels of the compacted domain, has a higher proportion of areas with more grain numbers than the non-compacted and the lower levels of the compacted samples. These results illustrate the non-uniform spatial distribution of the physical changes that the soil undergoes through compaction. The impact of compaction decreases with depth away from the source of compression.

Macroscale soil compaction changes were evaluated by analyzing images that captured the movement of colored sand grains and by measuring their translocation before and after compaction. The highest translocation was at the upper levels of the soil profile, and with depth translocation decreased. The distances between adjacent selected grains, before and following compaction, indicated that compaction is not uniform along the sand profile, with certain levels compacted more than others, strengthening the assumption of the heterogeneous nature of soil compaction and the shear band effect.

Since compaction affects the particle arrangement along the profile in a non-uniform manner, with maximal compaction at the relatively high layers of the soil profile, the impact of compaction on combined evaporation and salt precipitation was observed over two setups that were considered compacted domains: (i) layered columns packed with glass beads with increasing size with depth (FU), and (ii) coarse-texture sand samples which were manually compressed from

the top. In compression, three setups were considered non-compacted domains: (i) a homogeneous column packed with mixed glass beads, with a wide range of particles size distribution (HM); (ii) a layered glass-bead column where the glass-bead sizes decreased with depth, representing a tilled soil profile (CU); and (iii) a homogeneous coarse sand sample. The cumulative evaporation measurements and the visual observations pointed to the significant impact of the different configurations on combined processes of evaporation and salt precipitation.

Glass-bead experiments and numerical simulations have shown that for the HM structure the drying front recedes from top to bottom, as expected for evaporating homogeneous porous media. The relatively long S1 duration and high cumulative evaporation of the HM setup resulted in a notable precipitation of an efflorescent salt crust at its upper surface. The precipitated salt layer resulted in a sharp decrease in evaporation rate since hydraulic continuity to the surface is lost, and the slow process of vapor diffusion through the salt layer controls evaporation.

For the FU, the drying front propagated from bottom to top, as demonstrated both experimentally and by the numerical simulation. S1 duration of the FU was long for the saline and DI water, and the cumulative evaporation was high. In the glass-bead experiment, a prominent efflorescent salt crust was precipitated at the FU upper surface, due to the long S1 and resultant high cumulative evaporation. However, in contrast to the HM, even though a notable salt layer was observed, its impact on evaporation at the FU structure was moderated compared to its impact for the HM setup. This is attributed to the stronger capillary suction of the upper layers, at the FU structure, which pumps water from the underlying levels upwards, maintaining high saturation at the soil surface, which supports liquid water continuity from the soil to the evaporation front, at the salt crust upper surface.

For CU setup, a moderate recession of the evaporation front downward occurred over time like in the HM configuration, and loss of hydraulic continuity to the surface was achieved relatively early. Thus, the cumulative evaporation was low, salt precipitation was minor, and therefore negligible differences in evaporation between saline and non-saline conditions were observed.

Results of the sand column experiments were in good agreement with the glass-bead column experiments and the numerical simulations. This concurrence between the different experiments and the simulations further supports the model and research hypothesis, as being presented in Fig. 1, and supports the assumption that the layered FU structure was a good approximation of compacted soil conditions. Nevertheless, it is important to emphasize that in real field conditions soil textures, environmental conditions, and compaction processes are much more complicated than those examined in this work. Therefore, further studies under more realistic conditions are needed.

This work sheds new light on the impact that soil compaction, which is a common feature in arable lands, has on bare soil evaporation processes for saline and non-saline conditions. The insights gained from this study indicate that one may consider the use of different agricultural practices to control the degree of soil compaction to the benefit of the water regime in the root zone.

*Data availability.* Raw CT scans used for image analysis are available: https://doi.org/10.6084/m9.figshare.14773212.v1 (Nachshon and Goldberg, 2021a).

Raw data from evaporation experiments are also available: https://doi.org/10.6084/m9.figshare.14773149.v1 (Nachshon and Goldberg, 2021b) TS4.

*Author contributions.* NG-Y designed the experiments, with contribution from all co-authors, and carried them out. NG-Y prepared the article with contributions from all co-authors.

*Competing interests.* The contact author has declared that neither they nor their co-authors have any competing interests.

*Review statement.* This paper was edited by Insa Neuweiler and reviewed by two anonymous referees.

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

**Remarks from the language copy-editor**

CE1  The original wording was ungrammatical. Apologies, adding "that" during copy-editing only made things worse. I have rearranged this sentence again with a different meaning. Does this convey your meaning better?

CE2  Please confirm the changes here.

**Remarks from the typesetter**

TS1  Due to the requested change, we have to forward your request to the handling editor for approval. To explain the correction needed here to the editor, please send me the reason why this correction is necessary. Please note that the status of your paper will be changed to "Post-review adjustments" until the editor has made their decision. We will keep you informed via email.

TS2  Please confirm updated unit.

TS3  Please confirm updated unit.

TS4  Citations were added.

TS5  Please provide date of last access.

TS6  Reference was added.

TS7  Reference was added.

TS8  DOI does not seem to work.