# Peer review of "Compaction effects on evaporation and salt precipitation in drying porous media"

_Hydrology and Earth System Sciences, 2021_

## Author Comment (AC1)

**Reply to reviewer #1:**

The paper reports experimental results of evaporation and salt precipitation from heterogeneous (layered) soil columns. The motivation of the study is to quantify effects of soil compaction on evaporation losses and salt precipitation. This is an important topic to understand water and solute fluxes in arid regions and for agricultural practice with irrigation. As analog of a compacted porous medium, the authors use a layered column with increasing particle size with depth. They motivate/justify this analogy with experimental results of an imaging study using micro-tomography that indicates smaller sand particle sizes (and pore distances) after compaction (increasing pore size from top to bottom as in layered column).
The authors reported a large effect of layering on evaporation losses in column experiments, with longer stage-1 duration for columns with finest layers at the surface.
The authors reproduced these experimental findings qualitatively with simulations using Hydrus-1D simulations. The study contains many interesting elements (CT-imaging, lab experiments and simulations) but these elements are – from my point of view – rather poorly connected and need some additional analyses as explained in more detail in the comments below.

We thank the reviewer for identifying the importance of the study. Following the reviewer comments we are editing and rewriting parts of the paper, in order to strengthen the connection between the two parts of the paper, as detailed below. Another column experiment will be done, upon the reviewer comments below.

MAJOR COMMENTS

1) The paper contains two studies that must be better connected; one study is focusing on imaging the effect of compaction on pore and particle size characteristics and a second one on evaporation and salt deposition from layered columns. The connection between the two parts is given by the hypothesis, that compaction results in smaller particle and pore sizes close to the surface with a profile of increasing pore size similar to the chosen layering. From my point of view, it would be important to connect these two studies by measuring evaporation from non-compacted and compacted sand columns (without layering). Could the authors conduct such experiments (not with the same sample used for measuring particle displacement, but packed with the same compaction method)? With water and salt solution? These experiments could easily be conducted and connect the two parts of the study.

We agree with the reviewer that the two parts of the paper has to be better connected, and we liked the suggestion of the column experiments. We have finished one cycle of the proposed experiment and another repetition is currently running. The first repetition of the experiment showed very nicely that a "normal compaction", not in a layered structure, affects evaporation in the same way as observed for the constructed layered structure. These new experimental results will be incorporated into the revised manuscript.
In addition, we changed the introduction section to include information about previous similar column experiments that were done in other studies. The differences between these works and ours are presented, but in general, the cited works supports our hypothesis and results. This addition, we believe, will strengthen also the connection between the different parts of the paper.

For our own check - we conducted another simulation in HYDRUS for a domain with a gradual change of the soil hydraulic properties, and the results are in agreement with the experimental results and the simulation of the layered structure presented in the paper. In order to keep the paper simple, and since we consider the

simulation only as another tool to prove the concept proposed by the hypothesis, we refer not to add more simulations, under different setups, to the paper.

2) Layered columns as analog of compacted columns: my main concern is that conclusions on effect of compression (typically with sigmoidal or exponential profile of bulk density) are made using a layered system with a stepwise change in bulk density. In contrast to the macroscopic trend in bulk density reported in literature, the bulk density in the layered media is not decreasing with depth but the maximum bulk density (minimum porosity) was close to the bottom (see Table 1). Accordingly, the bulk density profile in the layered column is very different from the one expected for compaction. The expected trend of increasing pore size with depth may be partially represented by the chosen layering with increasing particle size with depth. However, the layering with sharp contrast of pore sizes will have other effects than a gradual change of bulk density (and pore sizes). For example, when the tip of the drying front crosses the boundary between fine and underlying coarse material, the capillary pressure jumps abruptly from a more negative to less negative value with rapid water redistribution from coarse to fine layer. Such abrupt changes are not expected for gradual changes in pore sizes. Accordingly, I don't know how representative a layered column is for a compacted column with gradual change of bulk density. Also from that point of view, the additional experiments proposed above would be important.

These are good points and we agree that the new column experiment will help to strengthen the notion that the layered structure is a good representative of a "normal compaction". As detailed above, the manuscript will be revised to explain this point and the differences between a layered structure, and a gradual change of the media properties.

3) Numerical study: in Figure 5, the authors show that the addition of a salt layer on top of a homogeneous column stops stage-1 evaporation; but for the layered column, stage-1 evaporation can be sustained. It would be important to show the pressure evolution at the surface (or in profile) as a function of time to make clear why the capillary pumping will stop when adding a 'loamy sand layer' (as analog of the salt layer) in case of the homogeneous column but not the layered column.

Will be added. Figures below presents surface pressure head over time for the compacted state (left) and the non-compacted state (right), with no salt precipitation. This shows very nicely the moderate fall of pressure head at the compacted setup, and the sharp increase in pressure head for the non-compacted state.

[Figure]

Underlying images present pressure head along the soil profile, for the saline state and it is seen that for the compacted set up (left), the pressure head is higher (less negative) than the non-compacted setup. Therefore, a hydraulic continuity

between the drying soil and the salt crust is being maintained at the compacted state. Figures like these will be added to the revised manuscript.

[Figure]

To be consistent with the other lab experiments, the same simulations should be conducted with (i) the reverse layering and (ii) a gradual decrease of bulk density from top to bottom.

As detailed above, the numerical model is not the lion share of the work and is presented only as a proof of concept and used to strengthen the hypothesis. Therefore, and in order not to make the paper too long and cumbersome, we prefer not to present any additional simulation results. We believe that the added column experiment and the addition of more theoretical discussion to the paper, will contribute more to the understanding of the reader.

4) Effect of salt precipitation: based on the numerical experiments, the authors hypothesize that the deposition of salt affects the evaporation stage dynamics differently for homogeneous and layered columns. In the numerical experiments, evaporation stage transition occurred just after salt deposition for the homogeneous column; however, for the lab experiments, stage transition occurs much earlier (~15 hours based on Figure 7) than salt deposition (54 hours, page 23, line 508). But for the heterogeneous column, the stage transition occurred soon after salt precipitation (at 60 hours based on Figure 7, six hours after salt precipitation that occurred after 54 hours, page 24, line 517). Accordingly, the observations do not fully correspond to the findings of the numerical study. The authors should comment on that.
The discussion of the salt precipitation in the lab experiments is not consistent. With respect to salt precipitation in the homogeneous column, it is stated on page 23, line 508, that precipitation started after 54 hours. But on page 28, lines 611-612 it is concluded the development of salt crust started after 10 hours. This is inconsistent. To agree with findings from the numerical studies, it is expected that a dry salt layer is built long before 54 hours (i.e. at end of stage-1). Do the authors have any experimental evidence (images) that salt was deposited after 10 hours?
For the heterogeneous column (FU), it is argued on page 29, lines 620-622, that the crust was hydraulically connected to the underlying medium. However, the stage transition (60 hours) occurred very soon after salt precipitation (54 hours, page 24, line 517) and it seems that liquid connections are not sustained very long through the salt crust. Accordingly, the conclusion that the hydraulic connection includes the salt crust could not be proven (page 30, lines 638-639). Or do the authors have experimental evidence (images) that there was salt deposition before 54 hours?

We regret for this unnecessary confusion. The revised manuscript makes order in these disparities.

5) CT analysis: for the quantitative analysis of compaction, only five cross-sections per region were used to estimate grain number and area and mean pore distance. From my point of view, the change in porosity or bulk density is very important as well because bulk density is the property that is measured macroscopically (as described in the introduction). The authors should conduct an image analysis that shows the profile of porosity and bulk density (using all layers, not just five cross-sections per region) to reveal the effect of compaction with depth.

The presented images are vertical transects and not horizontal cross sections (as understood by the reviewer). Therefore, we believe it gives a reliable information about the changes of the different properties, with depth. Apparently, at the length scale of the presented scans (~8 mm high) the changes are too small to be noticed. For this reason, we did the top scan (0-7 mm depth) and the deeper scan (9-18 mm depth), which allow to observe the differences in the soil physical properties between deep and a shallow samples.

In order to avoid similar misunderstood by the readers, this will be better explained in the revised manuscript.

6) Grain breakage: in Figure 2 and text (page 16, line 357; page 18, lines 406-407) the term 'grain breakage' is used, indicating that sand particles break during compaction. For me it is not clear that sand particles break based on the presented analyses. For example, did the authors check in 3D if the particles shown in Figure 2a''' were really broken or if it is just a different arrangement of particles?
The quality of the applied particle counting method cannot be assessed based on figure 2 – an inset with higher resolution would be needed to show the performance of the grain counting method.

We will add 3D images that show nicely the differences between compacted and non-compacted conditions. It is clearly seen that under compacted conditions there are much more particles, a result of the breakage. See images below.

As for Figure 2(C) – the intention was not to allow the reader to count the grains. The purpose was to present the "color map" analysis that we used, which enables to observe visually the number of grains at each block of the generated grid. We will rephrase and improve the text to make it more clear. In other words, the reader should be able to identify the different colors of the grid boxes, and not to count the number of grains within each box.

Bottom (not compacted):

[Figure]

Compacted:

[Figure]

7) Conceptual model: The conceptual model presented in Figure 1 and described in the text needs some clarifications: (i) the authors should explicitly explain the difference between the drying front and evaporation front and the corresponding motion;

The entire section was rewritten and the conceptual sketch was improved based of the comments of the two reviewers (see draft of the revised image below)

[Figure]

**Figure 1: Conceptual model of evaporation and salt precipitation under compacted and non-compacted conditions; (A) initial stages of evaporation with the first air entry into the matrix. (B) advanced stages of evaporation, with the receding drying front and the hydraulic discontinuity between the soil and the salt crust (yellow layer), for the non-compacted soil, and the hydraulic continuity between the salt crust and the underlying soil, for the compacted state.**

(ii) the air entrance into the deeper layers must be explained (How does the deeper layers become unsaturated? I expect that there is air invasion in the non-compacted subsurface due to a large pore in the compacted region that brings air to the coarse layer.) Based on Figure 1, it could be concluded that a different

deposition pattern causes different evaporation rates for the two columns and that the dry top soil layer is limiting the evaporation rate. However, based on the numerical model, the drying of the salt crust and its hydraulic properties in contrast to the soil properties define end of stage-1. This could be clarified in the captions.

The reviewer is correct. The air has to come from some large pores at the surface. It is explained at the revised manuscript and seen in the revised conceptual sketch presented above.

8) Evaporation stage transition: The point of evaporation stage transition is used in several analyses. How was this transition determined based on experimental data?

Change in evaporation stages was determined by identifying the change in evaporation rate, upon the measured cumulative evaporation. It was validated by comuting the derivative of the cumulative evaporation graphs. Will be explained in the revised manuscript.

MINOR COMMENTS

Page 1, line 25: because the rates are the same in stage-1, I propose to write "evaporation losses", not "evaporation rates" that are higher in presence of small pores.
DONE

Page 8, line 178: The authors should differentiate between motion of the evaporation front and drying front and discuss the displacement separately; otherwise, there may be some confusion. In this paragraph, the authors state that (i) in case of non-compacted soils the evaporation front moves downwards from the soil surface in transition to stage-2 and (ii) in case of compacted soils in stage-1 there is a reverse process with a continuous flow of water from the deeper layer to the surface. But both statements are true for both compacted and non-compacted layers: in stage-2, the evaporation front recedes from the surface for compacted and non-compacted soils and in stage-1 a continuous water flow from deeper layers to the surface is sustained for both columns. What is different, is the direction of the displacement of the drying front (interface between saturated and partially dry medium) that is downwards directed in non-compacted (and non-layered) soils but upwards directed in compacted/layered porous media.
We thank the reviewer for highlighting this point. It is important and this section was rewritten and clarified, as detailed above.

Page 10, lines 225 and 234: Reducing the sample height of 30 mm by 2 mm does not correspond to an increase of 10% in bulk density (a decrease by 3 mm would correspond to 10%)
Changed to ~7%.

Page 11, line 260: Why did the authors choose different compaction values for CT (93%) and column study (95%)?
The two columns reacted differently to the compaction process.

Page 11, line 264: How could it be ensured that the very same grains were found before and after compaction? Could the authors show an example how they could identify the same particle?
In fact, it was very easy to identify the grains by the naked eye. In the materials and method section we emphasized that it was a "visual analysis of the images". In the revised paper, at Figure 4 we added few arrows to demonstrate the movement of selected colored grains (see below).

[Figure]

Page 12, line 282/283: The authors apply HYDRUS to solve water flow based on Richards equation that requires a continuous liquid phase with water content larger than 0.00 – the process leading to water content 0.00 is not simulated with HYDRUS. The justification of residual water content of 0.00 is not convincing.

When evaporation is not involved, the residual water content of each soil is indeed the lowest water content that will be achieved by regular water flow processes. However, evaporation will take any soil (after sufficient time) to zero water content. Therefore, we used this value. In practice it is not a big difference from using the residual water content, as also for the residual state – there will be no flow of water when reaching the very low water content values. However – because we wanted to simulate evaporation with the ability to get to zero water content we used the value of 0. Similar approach was taken by Zhou, Šimůnek, & Braud (2021) and we added a note about that at the M&M section.

Page 12, line 287: It is stated that n equal to 3 is the highest value permitted in HYDRUS. I made many simulations in HYDRUS with n larger than 3. Accordingly, I do not understand that statement. (by the way: in Assouline and Narkis (2019) HYDRUS simulations were conducted with much higher n values (Table 1)). Please explain.

This is true, but for the current combination of the soil physical properties, which many of them were defined experimentally and we couldn't "play with", higher n values gave relatively high errors, at the order of 6% (Relative error in the water mass balance of the entire flow domain).

We added a sentence saying: " …..n was taken as 3 for the uniform layers, as it was the highest n value permitted by HYDRUS, while keeping the relative error in the water mass balance of the entire flow domain, at low values at the order of 1% and below."

Table 1: Please use greek symbols for water content; providing saturated conductivity values (and alpha and saturated water content) with so many digits could be misleading because the values are not known with such accuracy.
DONE

Page 13, lines 300 ff: The authors state that HYDRUS is only valid during stage-1 evaporation with hydraulic continuity along the entire soil profile and limit the analysis to stage-1. Interestingly, in Assouline and Narkis, 2019, HYDRUS was applied for a similar set-up after stage 1 as well because "it has been assumed that daily averaged S2 evaporation rates could be considered to be limited by liquid flow from deeper soil layers so that Richards' equation could still be applicable".

Since HYDRUS is valid only for simulation of liquid water flow (and not vapor flow), it is impractical to use it when vapor flow plays a major role in the transport of water (vapor) from the evaporation front to ambient atmosphere. Assouline and Narkis stated this point exactly (and not as understood by the reviewer).

Page 14, line 306 ff: The authors should also specify the initial conditions of the added salt layer
Added.

Page 14, line 318: From a permeability in the order of 4 Darcy (one digit accuracy) it is concluded that the saturated conductivity is 376.32 cm/day (five digits accuracy) – providing a conductivity value with so many digits is misleading.

True. Fixed.

Figure 3: How is it possible that there are rectangles with 0 particles? Are there rectangles of 1.06 x 0.73 mm size without grains?

We understand the confusion. It is because grains are counted according to the location of their center mass of gravity. Therefore, it might be that a specific rectangle would be with no grains centers in it, therefore it would consider it self to be with 0 grains. This will be explained in the revised manuscript.

Page 20, lines 439 and 440: The statement that maximal compaction was measured in top parts and is in agreement with micro-scale study is not convincing: the maximum delta_D was found in 30 mm depth, while for CT analysis the compaction was only measurable at top 7 mm.
We omitted the part of the sentence saying that it is in agreement with the micro scale analysis and only stating that "maximal compaction was measured in the top parts of the sample".

Page 24, line 521: Did the authors observe salt deposition within the column for CU?
No salt crust was formed. Very minor crystals were observed in a very limited loactions. It is now stated in the text.

Page 24, lines 529-531: Based on Figure 7, the cumulative evaporation dropped from 14 to about 2 mm for saline conditions. This is much more than 50% as was stated in the text. Please clarify.
A mistake. Changed to say "more than 80%". The source of the mistake is that it was compared to the cumulative evaporation of the DI column, at the same time and not at the time of S1-S2 transition.

Page 25, line 551: Do you mean the drying front? When the evaporation front is receding, the system is already in stage-2.
Changed to 'drying'.

Table 2: What are the units?
As original data – [mm]

Page 28, lines 594-595: S1 duration for DI set-up is about 65 hours (70 hours as stated seems a bit high) and this is quite shorter than 100 hours
Changed. This change is part of a wider edit that we did for this results section.

Figure 8: The definition of the change is strange; should it not be 'evaporation_DIevaporation_NaCl' divided by 'evaporation_DI' (multiplied by 100)?
True – Changed

Page 31, line 681: the drying front recedes from top to bottom in stage-1, not the evaporation front.

Changed

The paper discusses the impact of soil compaction leading to a vertical gradient in hydraulic properties and grain size distribution on evaporation from soils, on the formation of salt crusts, and on the feedbacks between salt crust formation and evaporation dynamics. An interesting aspect is that the formation of a salt crust can have an additional impact on the dynamics of evaporation from saline soils, in addition to the impact of the osmotic potential on the vapor pressure. Depending on the hydraulic properties of the porous medium and of the salt crust, either the evaporation is reduced instantaneously when the salt crust is formed or the evaporation may be sustained at a potential rate for a certain time after the crust was formed (an aspect that clearly comes out of the numerical simulations which could be highlighted more). The timing of the reduction depends on the hydraulic properties of the salt crust and the underlying porous medium. The paper discusses in detail how the vertical variations (or gradients) in hydraulic properties of the underlying porous medium play a role in the evaporation dynamics and salt crust formation.

One part of the paper presents the impact of compaction on vertical gradients in grain and pore size of a coarse sand. Two types of methods are used: micro CT and macroscopic images of colored sand particles from which particle displacements and changes in bulk densities are derived. The CT images clearly show the effect of compaction on pore and grain sizes and that the compaction effects are larger near the surface but vanish deeper in the soil sample. Effects of compaction are also visible in the macroscopic images but unlike what the authors write, I do not see convincing vertical gradients in compaction in these images.

We wish to thank the reviewer for the detailed and constructive review.

We agree that at the macroscopic analysis, the compaction trend is not as neat as seen by the micro CT analysis. However, as detailed in the paper, there is a general trend of high compaction at the top of the column with a reduced effect downward. Highest compaction is observed at depth of 30mm. We will improve the text to explain that at the macroscale, the impact of the "stress chains" is more notable compare to the micro scale, therefore – the changes are not as monotonous as observed by the CT scans.

A coarse sand is investigated and compaction of this coarse sand clearly led to a change in particle size due to a breaking up of sand particles. However, I am wondering whether compaction would have a similar effect on particle size in other texture classes (fine sand, silt, clay). Another aspect is that coarse sands are not known as soils where compaction has large effects on porosity. I would expect larger effects of compaction in silty and clayey soils. This could be addressed maybe in the discussion section.

This is a good point that deserves a note in the revised discussion section. Previous works have shown similar effect of compaction on evaporation, is fine texture soils. In these soils, it might be that grains breakage won't be seen, but reduction in pores size, will occur. Therefore – the overall effect would be similar. As aforementioned, it will be discussed and relevant works will be cited.

In a second part of the paper, simulation and lab experiments are carried out to demonstrate the effect vertical gradients in soil properties on evaporation dynamics from saline soils (including salt crust formation). The impact of compaction on evaporation is demonstrated in these sections using a layered porous medium with layers with different grain size distributions with larger grain sizes deeper and smaller grain sizes closer to the soil surface (information about the bulk density is not given). The layered profile is then compared with a uniform profile that consists of a mix of grain sizes. This mix of grain sizes should represent the non-compacted soil. But, I wonder whether this is a consistent representation of the non-compacted soil. To be consistent, the authors should have used a uniform system that consists of the grains from the lowest layer in their layered setup. By using a mixture of grains, they generate a porous medium with a wider pore size distribution (as is reflected in the lower n value) and also a lower porosity (as is reflected in the lower saturated water content) which therefore has also a higher bulk density. The comparison between the mixed-uniform and layered systems is therefore not a suitable analogue for a comparison between compacted and non-compacted soil. The mixed uniform soil represents a medium with a wider grain size distribution and pore size distribution. According to the first part of the paper, this should be a characteristic of the top compacted layer but not of the non-compacted soil. Therefore, I think that authors should best include a new experiment with a uniform sample that represents the non-compacted lower soil layer.

We agree. Unfortunately, we do not have enough glass beads of the coarse texture to perform the suggested experiment. However, responding to the comments of reviewer #1, we are currently doing a column experiment with homogeneous coarse texture sand (same sand used for the CT analysis), for compacted and non-compacted conditions. We believe this would be a reasonable solution for the point raised by the reviewer.

In general, the paper was well written. But, some parts should be described more clearly since readers might easily draw incorrect conclusions from certain sentences. For instance, the focus on the higher capillary suction that can be exerted by a fine top soil layer gives the impression that more water can be pulled up from coarser deeper soil layers than in case the fine textured layer is not present and that therefore more water can be evaporated when a fine textured layer is present at the soil surface. First, the authors should always make clear when they make comparisons with what they are comparing, i.e. what is the base case: the uniform fine texture layer or the uniform coarse texture layer. Often comparisons are made but it is not explicitly clear with what the comparison is made. In the list of detailed comments, examples of these comparisons are given.

The introduction, conceptual model, and discussion sections will be rewritten, to improve clarity and better explain the complex processes discussed in the paper.

 Second, the conclusion that more water can be pulled from underlying coarser layers when a fine layer is on top is incorrect (the authors do not write this but from what they write, one could easily draw this incorrect conclusion). The reason for larger evaporation losses when a fine layer is on top is that almost the same amount of water can be extracted from the underlying coarser layers and to this amount of water, the water that can evaporate from the fine layer can be added. The conceptual figure 1 does not really illustrate this but could be easily adapted to make this clear.

We are not sure we completely understand the reviewer comment. However – the entire paper was reedited and some parts were rewritten and we believe it would be more clear now. The conceptual model was better described and Figure #1 was changed to include the dynamics of drying front propagation (see comment #7, reviewer #1).

A second issue that should be explained better is the reason why the presence of the fine layer keeps the salt layer on top better connected to the deeper soil. The authors argue that it is related to the wetness of the fine layer that keeps the hydraulic connection. I think this should be rather the capillary pressure of the fine layer which is not too high (or pressure heads too low) so that the salt layer is not dried out too much and the water at the top of the salt crust stays hydraulically connected to the deeper soil. Therefore, the statement that the higher capillary pressure in the fine layer keeps the hydraulic connection between the salt crust and the deeper soil layers is to my opinion incorrect since it should rather be the opposite. In addition to showing the simulated wetness, I propose to include also simulated capillary pressure heads to make this clear.

This was raised by reviewer #1 also and it will be improved through deeper discussion in the text. It is true that for the compacted conditions, the fine layer is more wet, with lower matric pressure, therefore water flow into the crust is possible. This will be better explained in the revised manuscript, by presenting simulation results that include information about pressure head changes at the media / salt crust (see reply #3 to reviewer #1).

Specific comments:

Ln 25: ‚comprised' The wording was not clear to me whether 'comprised' means an increased connectivity or decreased connectivity.

Changed to:"….due to the rising of hydraulic conductivity….."

Ln 53: Isn't the reason for compaction after tillage that soil aggregates slake after rain and the slaked particles create a crust?

This is true for the soil surface. However, in this sentence we are discussing the " the soil at the lower boundary of the tilled zone ".

Ln 54: Isn't that the plough pan?

 Yes it is. We added this term to the sentence.

Ln 63: The uneven distribution of hydraulic properties does not necessarily lead to anisotropy. What determines the anisotropy is the shape, orientation of the heterogeneities.

Due to the lack of quantitative definition of the degree of samples anisotropy – we will omit it and use the term heterogeneity only.

Ln 111: manly à mainly

Done.

Ln 120: A precipitated salt layer may increase evaporation… : with respect to which conditions would the evaporation be increased? I am a bit sceptic that a salt layer can really 'increase' evaporation compared to the evaporation rates that would occur when there is no salt layer. Especially during phase I, when evaporation is mainly controlled by the available energy, I expect no big influence of the salt layer (even a reduction of evaporation since the albedo of the salt is higher than of the soil). I suppose the authors are referring here to the effect of the pore size distribution, which may be finer than in the underlying soil generating a higher capillary suction, on evaporation. The higher capillary suction in the top layer can lower the drying front in the underlying layer, but not more than by the thickness of the top layer. Thus, a thin top layer will lower the drying front only very little. The lowering of the drying front in the underlying layer by the thickness of the upper layer does not increase the amount of water that can be evaporated during phase I evaporation from the underlying layer. The increase in evaporation during phase I evaporation comes from the extra water that evaporates from the top layer. Thus, when this layer is very thin, there will be almost no effect on evaporation. This was also demonstrated by Li et al. (VZJ, 2020, DOI: 10.1002/vzj2.20049)

As explained in the text, this possible increase is a result of increase of the evaporating surface area, as the crystalized salt has a larger surface area compare to the underlying soil. For more information, please see Shokri-Kuehni et al., 2017.

However, we edited the sentence so it would be very clear that it might be correct in certain cases only and not always.

Ln 156: I agree that the higher capillary suction of the compacted top layer can pump up water from deeper in the underlying noncompated soil layer. But, this depth does not depend so much on the magnitude of the capillary suction that may be exerted (at least when it is above a certain threshold) by the compacted layer but rather on the thickness of the compacted layer. Furthermore, the depth from the top surface of the noncompated layer from which water is pumped up, does not increase. It is the depth from the soil surface that increases.

Dimensions of the compacted layer are important, but the fundamental property Is the ratio between the characteristic length of the porous medium and the thickness of the layer. In the revised manuscript we discuss the connection between the thickness of the compacted layer and the depth of the drying front, which are both affected by compaction.

Ln 181: 'where a continuous flow of water is sustained from the deeper layer of the soil profile to the soil surface, extending the duration of S1 and allowing more water to evaporate.' See comments above. First, it is important to explain with respect to

which condition the duration of S1 is extended. When it is with respect to a fully compacted soil layer, i.e. without an noncompated layer below it, then I think it is not correct to state in general that an noncompated layer below a compacted soil layer leads to an extension of S1. When it is with respect to evaporation from an noncompated layer, then the formulation could be read as if the extension of S1 by the presence of the compacted layer on the surface is due to the fact that more water is extracted from the underlying noncompated layer during S1. But, I think this would be a wrong interpretation since the presence of the compacted layer on top of the compacted layer cannot increase the amount of water that can be extracted from the noncompated layer. The only reason why more water can be evaporated during S1 by the presence of a compacted layer (in comparison to an noncompated soil) is because additional water is lost from the compacted layer.

We hope we understood the comment correctly. The compacted layer results in that more water from the entire media are being pumped upward and participate in the evaporation process (S1). Without the combination of compacted layer over a loose layer, this effect wouldn't happen. The entire paper was revised to make sure this point is clear to the reader.

Ln 183: 'Consequently, the deeper soil layers dry out first, while the upper layers remain at relatively high levels of water content (Figure 1B(2)).' I agree. But, if the upper layers do not lose water during S1, then S1 will not be extended and there would not be more salt precipitating. So, I think there is a conceptual problem with figure 1 because the same amount of water was lost (and therefore the same amount of salt should be precipitated) in figures 1B1 and 1B2. I propose to include also some air in the compacted layer so that water is lost from figure 1Bmore is lost.

Figure was changes to include the process of air entry (see image above at the reply to reviewer #1). In addition, the text of the conceptual model section will be improved also, in order to explain the important issue of air entry into the matrix.

Ln 187: 'In noncompacted conditions, the precipitated salt crust reduces evaporation as it acts as a barrier that reduces water vapor diffusivity from the evaporation front to the soil surface and to the atmosphere (Figure 1B(1)).' But this reduction occurs during S2 and would also occur during S2 in the compacted soil.

This section was rewritten, to better clarify that in homogeneous, non-compacted conditions, the salt crust is mostly with no hydraulic connection to the pore water, therefore it reduce vapor diffusivity. For compacted state, however, the liquid water flow through the salt crust is possible due to hydraulic continuity between soil pore water and the salt crust.

Ln 193: '…its impact on evaporation will be moderate compared to non-compacted conditions.' I think the impact is rather related to where evaporation is taking place, i.e. small impact when the evaporation takes place at the surface of the salt crusts but larger impact when the evaporation takes place deeper in the soil profile. Also

in the compacted soil, the impact of the salt crust might be large during S2 whereas its impact might be small in the noncompacted soil during S1.

As stated above. This is better explained at the revised manuscript.

Ln 226: 'with typical grain diameter of ~500 µm (sand characteristics can be found in Nachshon, 2016)' What does the 'typical grain diameter' represent: the median, mode, ….? I propose to include also a uniformity index of the grain size distribution.

This is the mean. Text was changed to say that.

Ln 245: Which Matlab libraries or toolboxes are these functions from?

Image Processing Toolbox. Added to the text.

Ln 260: Shouldn't it be 'increased' instead 'reduced'?

True. Changed.

Ln 295 table 1: I would propose to include the hydraulic properties of the salt layer also in this table. The parameters are related to the particle size except for the saturated water content. Maybe a reason for this different behavior of the saturated water content with particle size could be given.

The parameters were added to the table.

Ln 324: I think that also fluid viscosity and surface tension are influenced by the salt concentration.

The sentence was changed to say that these properties may be affected also by the changes of the solution salt concentration.

Ln 360: grain sizes? Shouldn't that be grain numbers?

True. Changed.

Ln 400: Figure 3: grain number, grain area and mean pore distance should have units. I do not understand the relation between grain number in Fig 3A and grain number in Figure 3B.

In Figure 3A, all parameters are normalized in respect to the non-compacted sample (as explained in the figure caption and the text). Therefore, it has no dimensions.

Figure 3B is not derived from Figure 3A. grains number in Figure 3A presents average of number of grains in a given surface area for the selected images. Then all averages were normalized in respect to the non-compacted state. In 3B however, the histograms present the number of the squares in the matrices (From Figure 2C).

This is explained in the text of the revised manuscript.

Ln 438: 'Nevertheless, it is evident that most of the profile underwent compaction, as most of the Δð• • · values are negative, and that maximal compaction was measured in the top parts of the sample, in agreement with the results from the micro-scale study.' I do not really see this in figure 4B. I would rather say that the compaction is heterogeneous but does on average not differ a lot with depth.

In accord with reviewer #1, the sentence was changed to say: " This analysis further emphasizes the heterogeneous nature of soil compaction and the shear band effect. Nevertheless, it is evident that most of the profile underwent compaction, as most of the ∆D values are negative. Maximal compaction was measured in the top parts of the sample, at depths of 10 and 30 mm, but a notable compaction was measured also at depths of 65 and 80 mm."

Ln 454: '…including the unique pattern of drying from top to bottom,' Shouldn't it be reverse: unique drying from bottom to top?

True. Changed.

Ln 474: 'the presence of the salt crust resulted in hydraulic discontinuity between the saturated lower parts and the upper surface of the domain.' I do not follow this statement. Looking at figure 5A2, the water content in the soil profile is quite uniform with depth so that the lower parts of the domain are not much more saturated than the upper parts of the soil column. It seems to me that the presence of the salt layer leads to a reduction of the effective hydraulic conductivity of the layered medium that consists of the salt layer and the underlying uniform soil.

Entire discussion of the model was rewritten

Ln 480: 'The fine media at the top of the FU profile maintained wetness conditions that enabled liquid water flow from the soil into and through the salt layer, to replenish evaporation at its upper surface.' I think the crucial point here is the capillary suction (and not the wetness) in the upper layer since that defines the suction and conductivity of the salt layer. I suppose that in the layered profile, the capillary suction at the top of the profile was lower (water pressure head was less

negative) at the time of the initiation of the salt layer than in the uniform profile. As a consequence, the conductivity of the salt layer would be larger in the layered profile than in the uniform profile so that it could sustain S1 evaporation longer.

Entire discussion of the model was rewritten. Some more discussion about the pressure head profile was added.

Ln 565: please check the figure caption. The labels do not correspond with what is shown in the figure.

True. Changed.

Ln 575, table 2. The standard deviations should have units.

True. Added

Ln 639 'We suggest that, in the case of a homogeneous soil, the receding evaporation front breaks the hydraulic continuity to the salt crust.' I suppose that this depends on the hydraulic properties of the homogeneous soil and not on the fact whether the profile is layered or homogeneous. What would be the difference between DI and saline solution evaporation when the homogeneous soil would consist of the fine soil layer material?

From previous works it is known that under homogeneous soil conditions, at various textures, the salt crust usually act as a mulching layer that reduce evaporation. In most cases the salt crust is not hydraulically connected to the soil, as stated in this work, therefore it limits evaporation. The introduction, conceptual model, and discussion sections were thoroughly changed to better understand the findings of this work and the fact that was seen that compaction support a better hydraulic continuity between the soil and the salt crust.

Ln 690: 'This is attributed to the stronger capillary suction of the upper layers, at the FU structure, which pumps water from the underlying levels upwards, maintaining high saturation at the soil surface' I rather think it is the opposite (but I am not sure and it would be helpful to show simulated capillary pressures or pressure heads). The fine layers on top keep the capillary suction for a longer time at a relatively low level so that the salt layer does not dry out and connectivity between the evaporating surface at the top of the salt layer and deeper in the soil profile is not lost. In the uniform soil layer, the capillary pressures increase more (pressure heads become more negative) so that the salt layer dries out earlier and the hydraulic connection between the top surface of the salt layer and the underlying soil is lost earlier and evaporation reduced earlier.

Entire section was changed.

---

## Author Response (AR2)

Dear Editor,
We wish to thank the two reviewers for their positive, thoughtful and constructive reviews. Below you will find our replies (in red) to the general and specific comments raised by the reviewers, and detailed description of the corresponding changes in the paper. Since the paper has been changed a lot from its original version, it went through another professional English editing, to ensure an easy reading. We hope you and the reviewers will find the paper suitable for publication.

The authors have made a substantial effort by including extra simulations and experimental data in their revised paper, which is highly appreciated.
Nevertheless, I think there are still a few issues that should be addressed.

First, the authors compared evaporation from layered (or top soil compacted) soils under saline and non-saline conditions with those from other soils and use these comparisons to draw conclusions on the effect of layering (or compaction) and salinity on evaporation. However, there are two issues the authors should pay attention to. First, they should always make clear with what they are referring, i.e. they have to clarify clearly the reference case. Second, they should define the reference case so that it is representative for the non-compacted soil. I do not agree that the case they are considering now as the reference case, namely the homogenized soil which has a much wider grain size distribution, than the layered soil that is used to represent a compacted soil, can be considered as a proxy for the non-compacted soil. Therefore, although the comparisons for the two cases: layered soils with layers of uniform grain sizes versus homogenized soil with mixed grain sizes is very interesting and illustrative, it does not represent a comparison between a compacted versus a non-compacted soil. Thus, it is not correct, to my opinion, to draw conclusions about the difference of evaporation between compacted and non-compacted soils based on this comparison. The comparison between the layered FU and CU soils is already interesting on its own and I am wondering what the additional value of the homogeneous HO scenario in fact is. To my opinion, it represents the behavior of another porous medium with a much wider pore size distribution. One could consider leaving it out and replacing it by a numerical simulation but using a homogenous soil profile that corresponds with the properties of the lowest layer (although this might be a bit extreme in hydraulic property contrast between the top and bottom).

This is a very important comment – thank you. We added another two setups to the model section: homogeneous coarse sand and homogeneous coarse sand with a 1 cm layer of fine matrix on top, to mimic compaction of the very top layer of the medium. The corresponding sections in the M&M (Section 3.3) and at the results (Section 4.3) where changed accordingly. Moreover, we went carefully over the text and make sure that the correct terms are being used to describe the compacted and the non-compacted scenarios.

Second, improve the quality of figure 7.
Figures 6 and 7 were changed and their quality was improved, in the revised section.

Third, explain in figure 11 why stage 1 evaporation of the saline compacted sand soil column is much smaller than the stage 1 evaporation of the saline non-compacted sand column.

We wonder if there may have been a mistake in the comment, since stage 1 evaporation of the saline compacted sand soil is longer than the saline non-compacted as expected.

Detailed comments:

Ln 119: 'It was shown that porous media composed of a fine texture domain that overlies a coarse texture domain may result in longer duration of S1 and increased cumulative evaporation.' As I commented in my previous review, it is important to mention the reference case. With respect to what does this layered medium evaporates more? With respect to a homogeneous coarse domain or a homogeneous fine domain? I think it is in most cases correct to state that the layered profile (with fine on top) will have more S1 evaporation than the uniform coarse profile. The depth of the drying front (i.e. where the transition between saturated and unsaturated conditions take place) in the coarse layer will be translated downward by approximately the thickness of the fine layer (if the thickness of the fine layer is smaller than its characteristic length) so that the same amount can be extracted from the coarse layer during S1 evaporation in the layered system as in the homogeneous coarse soil. Since water is also extracted from the fine layer on top, the total amount of water that can be evaporated from the layered system will be larger than the amount of water that is extracted from the coarse layer. Whether the layered system with a fine layer on top will evaporate more during S1 than a uniform fine layer, depends on the characteristic lengths of the fine and coarse soils, the thickness of the fine layer, and on the initial water content (or porosity when evaporation of a saturated soil is considered). When the thickness of the fine layer is much smaller than its characteristic length and when the stage 1 evaporation from the uniform coarse soil is lower than that from the uniform fine soil, the layered soil will evaporate less during stage 1 than the uniform fine soil. When the thickness of the fine layer is larger than its characteristic length, the layered soil will evaporate the same amount of water than the uniform fine soil. When the thickness of the fine layer is close to its characteristic length but smaller, then extra water can be wicked from the coarse layer below the fine layer at relatively low capillary pressures. Of course, this extra wicking is only possible when the water reservoir in the coarser layer did not drain by gravity (e.g. when there is a shallow water table).

Thank you for the detailed comment, it was added that we compare it to the initial case of a homogeneous coarse texture domain (page 5, line 121).

Ln 125 'Consequently, the coarse texture layer acts as a water reservoir that supplIES extra water to sustain a longer S1 and higher cumulative evaporation, compareD to a homogeneous FINE? soil structure.' See comment above. Can't it be stated that the thickness of the fine layer should be larger than Lcfine*delta_theta_fine – Lccoarse*delta_theta_coarse? In the line below, you mention that the fine layer

should not be thicker than a certain threshold, which I understand. But, I think it should also not be thinner than another threshold.

We are not sure we fully understand the comment. For the discussed situation of a fine texture domain over a coarse texture domain, the key issue is the ratio between the thickness of the layer and the characteristic length characterize the fine texture. This is explained in details in the revised version: "*It was shown that porous media composed of a fine texture domain that overlies a coarse texture domain may result in longer duration of S1 and increased cumulative evaporation with respect to homogeneous domain, composed of the coarse texture matrix only. In the layered structure, as soon as the drying front reaches the layers with the relatively larger pores, a rapid water displacement will occur from the large pores to the overlying finer pores. The pressure in the coarse layer changes abruptly from its air-entry value to the air-entry value at the evaporation front, which is associated to the higher capillary suction of the small pores (Or et al, 2007; Shokri et al, 2010). Consequently, the coarse texture layer acts as a water reservoir that supplies extra water to sustain a longer S1 and higher cumulative evaporation, compared to the homogeneous soil structure. It is important to emphasize that this process will occur only if the thickness of the fine texture layer is shorter than its characteristic length as only at this state the drying front may reach the coarse texture domain, while the system is at S1 and the evaporation front is still at the soil surface (Assouline et al., 2014; Assouline and Narkis, 2019)*."

Ln 152: 'In some cases, if the precipitated salt layer over the soil surface is hydraulically connected to the solution in the pores below, it may accelerate evaporation, as the surface area of the precipitated salt is usually higher compared to the underlying bare soil. Consequently, as long as the salt crust can pump liquid water from the underlying media, the elevated surface area of the salt crust would increase total evaporation' Why would an increase in surface area increase the evaporation? I think this would be related to the increase in surface roughness which increases for the same wind speed the latent and sensible heat exchange. This would imply an increased S1 evaporation.

As explained in the paper – increased surface area by the precipitated salt may increase evaporation as more area is available to contribute water for evaporation. It is explained in the paper and relevant papers are cited. However – we agree that salt crust roughness may also affect evaporation and we mentioned that at the revised paper (page 7, line 158).

Ln 218: compareD to…-
Done.

Ln 225: The salt crust will reduce (skip s) –
Done.

Ln 338, layer 60-80 mm: why does this layer have such a low porosity. Also the mixed layer has a much lower porosity than most of the layers in the layered soil profile.

It was determined experimentally, as detailed in the text.

Ln 523: compareD to –
Done.

Ln 550: I am still not convinced that a comparison between a simulation of a layered porous medium consisting of layers with a different textures and a porous medium that represents a mixture of the grain sizes of these layers is appropriate for a comparison between compacted versus non-compacted soils. This homogeneous layer would not represent a non-compacted soil.
I think the layered medium should be compared with a homogeneous profile that either consists of the layer with the coarsest texture or the layer with the finest texture.

Please see our reply to the major comment above. As said, we added a numerical model of homogeneous coarse texture domain and a compacted domain with a 1 cm layer of fine texture layer at its upper level. Now the paper covers four compaction / non-compaction setups, which are better explained in the text.

Ln568 Figure 7. The figure is hardly readable since the legend of the figure does not explain all the lines shown in the figure.

As detailed above – both figures: 6 and 7 were changed with the revised numerical model section.

Ln571: I do not understand which simulation results are shown here. When the evaporation is still in s1, the evaporation flux at the top of the soil and the interface with the salt crust should be nearly equal to the potential evaporation rate. But it is much smaller in the simulation. How can these simulation results then still represent S1 evaporation?

Entire section was revised.

Ln 666: 'no salt crust was observed (in CU) because of the low cumulative evaporation' I don't think that the low cumulative evaporation can explain this since the cumulative evaporation was nearly as high in CU as HO. In HO, a salt crust was built up. I think the reason is mainly the fact that in HO, the evaporation front remains longer at the soil surface so that all salt accumulation occurs there.

We agree and we changed the text to say: "*As aforementioned, for the CU saline conditions, no salt crust was observed because of the receding evaporation front, that did not support the processes of salinity buildup at the soil surface*".

Ln 747: 'The differences in the impact of salinity on evaporation between the HO and FU setups (Figures 9-10), together with the differences in patterns of drying (Figure 8), support the research hypothesis that even though more salt accumulation on the surface is expected in compacted conditions, its impact on evaporation is expected to be moderate compared to neutral conditions, since the hydraulic connection to the surface persists longer and includes the salt crust.' The problem is that the HO setup

must not be considered as the 'neutral condition'. The HO is a different porous medium than the FU and does not represent the 'uncompacted analogue' of FU. The closest that comes to the uncompacted analogue is the CU setup.-

As detailed above – this was corrected throughout the entire manuscript.

Ln 775: 'an experiments' Skip 'an.'
Done.

Ln 785: 'For saline conditions the compacted sand also displayed higher cumulative evaporation compared to the uncompacted state, with total cumulative evaporation of about 16.5 mm and 14.5 mm, for the compacted and uncompacted samples, respectively (Figure 11).' Under saline conditions, the cumulative evaporation is indeed larger for the compacted sand than for the uncompacted sand. But why is the stage 1 evaporation in the compacted sand much lower than in the uncompacted sand?

We afraid we didn't understand your comment correctly, the compacted condition has longer s1 as we were expected.

**REVIEWER #1**

Page 5, Line 118 and Line 125: Shokri et al. (2010) may fit better than Or et al. (2007) because the evaporation experiments from the layered sand was discussed there in more detail and more quantitatively.

Thanks - added.

Page 7, Line 154: It is an effect of enhanced surface area or reduced pores size (increasing the absolute value of the critical capillary pressure)?

It may be both, however, the reduced pores sizes in the salt surface are not yet commonly tested in the literature, and in this section we review previous studies.

Page 16, Line 371: "for", not "or"
Done.

Page 17, Lines 386-389: The sentence is a bit unclear; please be more specific.

The sentence was revised.

Page 18, Line 411: Can you specify "every few hours?"
Time gaps between the measurements were not consistent, due to technical constrains. Therefore, we used this term.

Page 35, Lines 725: The hypothesis that salt deposition resulted in hydraulic discontinuity and existence of 'mulching layer' for the HO-Salt experiment is – from my point of view – supported by the observation that the evaporation rate (based on Figure 9) is relatively constant between 20 and 100 hours (after S1) – it looks like the evaporation plane remains at the bottom of the 'mulching layer' for almost 100 hours; this is sort of 'secondary stage-1' with constant evaporation rate but with the evaporation rate below the surface at constant depth

This is correct. Text was revised to say: *"The precipitated salt crust acts as a mulching layer that results in hydraulic discontinuity between the saturated domain and the atmosphere. Even though the matrix under the salt crust is moist enough to support S1, under salt free conditions, it is not wet enough to support liquid water flow into the salt crust, therefore vapor flow through the salt dictates the evaporation rate. This is in agreement with observations from previous studies (Gran et al., 2011; Nachshon and Weisbrod, 2015), and the numerical simulation.* "

Page 38, Line 790 ff: The relevance and context of this very general statement is not clear; it could be explained in more detail in the following section 5.

We agree. This sentence was deleted and in section 5 we elaborated on that (page 39, line 854)

Page 39, Line 831: The authors mention several times that the drying front from compacted soil propagates from bottom to top. This is true in the case of extreme layering as used in the experimental study but in case of more gradual compaction it is possible that water is extracted continuously from entire profile but with higher water content close to the surface (see Figures 6 and 7: the water content drops in each time step in all layers, not just from bottom to top; there is no clear drying front moving from bottom to top); but in contrast to the HO case, the water content increases in upwards and not the downwards direction. The authors may consider to rephrase and explain in more detail.

We went over the text and better explained this issue.

Figure 3: The scale used for the distance map in (b) is not very illustrative and the effect on pore size cannot be seen; consider to use log scale or truncate the scale. In the captions you may state that these are vertical cross-sections

We tried several options and that was the best one. We believe that together with figs (a) and (c) – the message of the figure is clear.

Figure 4: Could you use the colors consistently in a and b (green is UN in (A) and LC in (B)).

It was a mistake in the legend. It was changed.

Figure 5: Should the curves in (B) not go to 0 for a column of 100 mm length?

Right, there was an error in the Y axis limits. Changed.

Figure 7: You may add in the captions that salt layer is 2mm deep and mark it by arrow in the figure.

Figure was changed and it present the hydraulic conditions prior to the salt crust onset.

Figure 8: Switch (A) and (C) in the captions;-
Done.

in (A) the homogeneous case is shown. You may state in (B) that there must be an air conduit "through the saturated layers" (in that sense they are not entirely saturated)
– added in line 619.

Figure 10: I propose to denote CU as "tilled" (not loose) according to the definition in text
 – Done.

---

## Author Response (AR3)

Dear Editor and reviewer,

We grateful for the positive review and the thoughtful comments. Here below we replied in details for all the comments. In general - comments (1), (2), and (4) were addressed in the revised manuscript and corresponding changes are described herein. However, we do not agree with comment (3) as explained below.

We wish to thank you again for the helpful review process, which greatly improved the paper and we hope that now you will find it suitable for publication at HESS.

The authors addressed most of my comments satisfactorily. There are two comments they seemed to have misunderstood. Finally, I noticed some problems with the scale of the setups that were used to investigate the length of stage 1 evaporation.

(1) Considering hydraulic soil parameters of the materials, the change of state 1 to stage 2 evaporation seems to be determined by the no-flow bottom boundary condition of the relatively short columns. Using a very simple steady-state solution of the Richards equation, I calculated that an evaporation rate of 0.65 cm/d can be sustained in the coarse material when the water table is at roughly 50 cm depth and at 190 cm in the homogeneously mixed soil. (Here I assumed that the tortuousity parameter of the Mualem van Genuchten unsaturated hydraulic conductivity curve was 0.5. The authors should include the value they used in their simulations). Based on this calculation, the duration of stage 1 evaporation should be much longer in the mixed soil than in the coarse soil whereas the simulation results in figure 6 show the opposite. This simple calculation shows that the duration of stage 1 evaporation in the 10 cm long columns must have been influenced by the no-flow bottom boundary condition. The question is how it influenced the simulations and experiments with the compacted layer and salt layers at the surface.

In the caption of Table #1 we added information about the tortuosity parameter, which indeed is equal to 0.5.

We agree that the length of the experimental columns and corresponding simulations were shorter than the capillary length of the different porous media. And indeed, if the profiles were longer they would have supported a longer S1. Nevertheless, we don't believe that the lengths affected the general behavior of evaporation and salt precipitation patterns under compacted and non-compacted conditions. We tested it numerically with a simulated columns of 100 cm long, and as expected, the length of S1 was extended. However – also for the long column setup – compaction (fine texture layer at the soil surface) resulted in longer S1, compared to the non-compacted setup. In order not to further encumber the text we prefer not to add more modeling to the paper. However, as suggested by the reviewer, we elaborated on that issue in the text and emphasized that the long columns experiments (Figure 11) tackle this issue [P18, L423-429].

(2) My comment is on Figure 11 was apparently misunderstood. The evaporation rate during stage 1 is clearly smaller for the compacted salt treatment than for the others. The slope of the cumulative evaporation versus time curve is clearly lower during stage 1 than for the other

cases. It is clear that stage 1 lasts longer for the compacted salt than for the uncompacted salt treatment. But the slopes of the curves for the two treatments during stage 1 is different. Under the same conditions, I would expect these slopes to be the same, and smaller than the slopes of the cumulative evaporation versus time of the DI water treatments (because of the lower vapor pressure of the salt water). In Figure 9, this is clearly the case but not so in Figure 11. Are the lower evaporation rates during stage 1 from the compacted salt treatment caused by the mulching effect of the salt layer that emerged on top of the compacted layer? But, if the salt layer reduces the evaporation rate, can we still speak of phase 1 evaporation? Or would it be a pseudo phase 1. I propose to make that clearer in the text why the slopes of the curves at the beginning of the experiment during the so-called phase 1 are different from each other.

This is an important point – thank you.

This reduced evaporation rate is not associated to flow resistance of the salt crust (for water or vapor), as no salt precipitation was observed during S1. In the revised section we pointed on this reduction in evaporation rate during S1 and proposed a mechanism to explain it:

*"S1 duration of the compacted saline setup was approximately twice as long as the non-compacted saline domain (~40 hour vs 20 hours, respectively). However, cumulative evaporation during S1 was only ~40% higher for the compacted setup (~4.8 mm) compared to the non-compacted domain (~3.5 mm), a result of lowered evaporation rate during S1 for the saline-compacted setup, compared to all other setups. Future studies should clarify this disparity, yet it could be related to preferential water flows that may be developed at compacted soils (Zhang et al., 2018; Shein et al., 2003; Nagy et al., 2018), due to heterogeneous changes of the compacted soil texture and structure along the stress chains (Nawaz et al., 2013; Naveed et al., 2016). Consequently, evaporation may be concentrated in specific locations at the soil surface and not homogeneously distributed as common for homogeneous domains. This may lead to elevated salt concentration at the pore water in the locations of evaporation, in higher levels compared to homogeneous distribution of the water at the soil surface, as demonstrated by Shokri-Kuehni et al. (2020). The increased salt concentration results in increased osmotic pressure and reduced vapor pressure of the solution, which reduces evaporation. It is likely that at the glass beads experiments this phenomenon was not observed since the glass beads layers were homogeneously packed. "*

(3) The second comments that was not fully understood is related to the text on Ln 117-ln 128 'In addition, structural changes of the soil along the vertical axis (with depth), may also affect evaporation (e.g., Or et al., 2007; Lehmann et al., 2008; Shokri et al, 2010; Assouline et al., 2014; Assouline and Narkis, 2019). It was shown that porous media composed of a fine texture domain that overlies a coarse texture domain may result in longer duration of S1 and increased cumulative evaporation with respect to the homogeneous domain, composed of the coarse texture matrix only. In the layered structure, as soon as the drying front reaches the layers with the relatively larger pores, rapid water displacement will occur from the large pores to the overlying finer pores. The pressure in the coarse layer changes abruptly from its air-entry value to the air-entry value at the evaporation front, which is associated to the higher capillary suction of the small pores (Or et al, 2007; Shokri et al, 2010). Consequently, the coarse texture layer acts as a water reservoir that supplies extra water to sustain a longer S1 and higher cumulative evaporation, compared to the homogeneous soil structure'.

This text block is still confusing to me. In the first part, the compacted layer on top of the coarser layer is compared with evaporation from a homogeneous coarse layer. The fine layer will indeed increase the duration and amount of evaporation but this increase is only due to the amount of water that can evaporate from the top fine layer in addition to the amount of water that can evaporate from the coarse layer below. I dispute that capillary forces exerted by the fine layer on top of a coarse layer will pull out more water from a coarse layer below than a free air flow over a coarse layer. Compared with the uniform coarse soil, the moisture profile is simply shifted downward (see figure 7A) so that at the same depth, the coarse layer below the fine layer is indeed drier than the uniform coarse layer. But, there is not more water extracted from the coarse layer. The only reason why more water evaporates from the compacted setup is because there is additional water that can be extracted from the fine layer on top. The presence of the fine layer on top of the coarse layer will not increase the amount of water that will be lost from the coarse layer compared to evaporation from a homogeneous coarse layer. Thus, again, the amount of water that can evaporate during stage 1 from the layered soil is only larger compared to the homogeneous coarse soil because there is extra water that can evaporate from the fine soil layer on top of the coarse layer. But, the second part of the text suggests that more water will evaporate since 'the coarse layer acts as a water reservoir that supplies extra water to sustain a longer S1 and higher cumulative evaporation, compared to the homogeneous structure.' When we keep the uniform coarse layer as the reference homogeneous structure, this reasoning is incorrect since there will not be more water extracted from this coarse layer whether a fine layer is present on top or not. The same 'water reservoir' is present in the coarse layer when no fine layer is on top. So the problem is that the authors are shifting the reference case and are now comparing evaporation from a layered system with a fine layer on top and a coarse layer below with evaporation from a homogeneous fine layer.

We do not agree with this comment. A key point of the paper is the extension of S1 by the fine texture layer that overlies a coarse texture domain. This setup forces the suction of excess water from the coarse texture domain towards the evaporation front. A lot of effort was put in supporting this concept both by experimental work and the numerical model. This concept is also supported by previous robust works which are being cited in the paper (e.g., Or et al., 2007; Lehmann et al., 2008; Shokri et al, 2010; Assouline et al., 2014; Assouline and Narkis, 2019).

It is well demonstrated in the paper that the fine pores layers on top of the coarse pores layers result in extension of S1 (Figures 6, 9 and 11), and the numerical model (Figure 7) and the column experiments (Figure 8) nicely support the concept of upward water flow from the coarse texture domain towards the fine layers near the soil surface.

Therefore, we prefer to keep on this message and to stand behind our statement as it is written in lines 117-128, and elsewhere in the text.

(4) Figure 7: Isn't the legend of the figure mixed up. Isn't black homogeneous mixed, blue full homogeneous coarse and dashed blue homogeneous coarse compacted?

True – Corrected.

---

## Author Response (AR4)

Dear Editor, Prof. Insa Neuweiler

Attached are the final version of the paper and the figures (zipped).

We wish to thank you and the two anonymous reviewers for the constructive review process and the acceptance of the paper for publication in HESS.

With best regards,

Uri Nachshon.